# MGAL: A Multilingual Granularity-Aware Long-Context Benchmark

**Chunhan Li** [1 2]   **Chenglin Xu** [2]   **Zongyang Zhang** [2]   **Jiale Liu** [2]   **Zhuoxi Rao** [3]   **Xudong Jia** [2]   **Junxiu He** [2]   **Menglin Yang** [1]   **Wenjuan Gong** [2]   **Zhengzhe Liu** [4]   **Chengwei Qin** [1]

## Abstract

Evaluation of long-context Large Language Models (LLMs) has advanced rapidly. However, most existing benchmarks are limited to the document level and focus mainly on high-resource languages, leaving many fine-grained challenges insufficiently evaluated. To address this gap, we present **MGAL**, the first multilingual, granularity- and position-aware long-context benchmark. MGAL is constructed from United Nations (UN) reports spanning 8K to 128K tokens across the six official UN languages. It covers four coherent levels of linguistic granularity (word, sentence, paragraph, and document) and further stratifies entries by their position within the document (begin, middle, and end), indexed at both the document and paragraph levels. This design enables systematic diagnosis of multilingual long-context comprehension across different granularities.

Through extensive experiments and analyses, we find that: (1) LLMs perform well at word-level tasks but struggle with coarser-grained ones; and (2) Closed-source models retain a clear performance advantage in lower-resource languages. We further identify two new challenges: (1) Under local semantic crowding, where neighboring sentences share topics and entities, models tend to follow surface cues (e.g., connectives like "however" or repeated entities) rather than the discourse role of the sentence in surrounding context (e.g., background, outcome); and (2) A gap between fluency and consistency in generated outputs, where models produce text that reads smoothly but drifts from the source facts. In addition, we observe several patterns in line with prior studies, including reliance on nearby evidence and reuse of options under uncertainty.

## 1. Introduction

Large Language Models (LLMs) have achieved remarkable progress across a wide range of Natural Language Processing (NLP) tasks. An important frontier lies in long-context modeling, where LLMs are required to process books, reports, and other documents spanning from thousands to hundreds of thousands of tokens. Effective comprehension of such long-form inputs is essential for applications like summarization, knowledge-intensive QA, and policy analysis. However, it remains highly challenging, as models must capture fine-grained cues across multiple discourse levels while maintaining robustness to positional variation.

To measure progress, several benchmarks have recently extended evaluation beyond short inputs. LongBench assembles a multi-task suite for long contexts (Bai et al., 2024), M[4]LE expands task and language coverage (Kwan et al., 2024), LV-Eval explores long-sequence tasks (Yuan et al., 2024), and ONERULER increases multilingual coverage (Kim et al., 2025b). While each of these benchmarks provides valuable advances, they also share important limitations: evaluations primarily focus on document-level in higher-resource languages, offer limited control over the positioning of evidence, and rarely examine fine-grained understanding across different discourse units. This gap highlights a fundamental question: how well do LLMs perform across varying levels of granularity in long-context, particularly in lower-resource languages?

To answer this question, we introduce MGAL, the first multilingual, granularity- and position-aware long-context benchmark, sourced from United Nations Digital Library reports of 8K–128K tokens across the six official languages. MGAL partitions tasks into closed-ended and open-ended groups. Closed-ended tasks admit clear, objectively scorable answers; open-ended tasks allow multiple acceptable responses (e.g., paragraph filling, summarization). For every language, MGAL covers four levels of linguistic granularity with seven tasks and 420 query–response pairs each (table 1),

[1]The Hong Kong University of Science and Technology(Guangzhou) [2]China University of Petroleum (East China) [3]Northeastern University at Qinhuangdao [4]Lingnan University, Hong Kong. Correspondence to: Chengwei Qin <chengweiqin@hkust-gz.edu.cn>.

*Proceedings of the 43$^{rd}$ International Conference on Machine Learning*, Seoul, South Korea. PMLR 306, 2026. Copyright 2026 by the author(s).

*Table 1.* Overview of MGAL task design across four levels of granularity in six UN official languages.

| Granularity | Task | Avg Len | Metric | #Data |
|---|---|---|---|---|
| Word | Single-QA | 23,887 | Acc. | 420 |
| | Multi-QA | 29,048 | Acc. | 420 |
| Sentence | Cloze | 30,838 | Acc. | 420 |
| Paragraph | Filling | 33,687 | Rouge-L | 420 |
| Document | Summarization | 28,189 | Rouge-L | 420 |
| | Translation | 33,294 | BLEU | 420 |

including word-level QA, sentence-level cloze, paragraph-level filling, and document-level summarization and translation. All items are annotated from scratch to broaden coverage of domains, input lengths, languages, and task types. Beyond granularity, MGAL incorporates position awareness by stratifying examples according to evidence location (beginning, middle, end), indexed at the paragraph and document levels. Following data construction, we prioritize quality over quantity by manually auditing every query–response pair and removing flawed samples.

MGAL uses position-aligned UN documents, where same-position sentences are semantically matched across six languages, enabling consistent cross-lingual comparison. Building on this alignment, we evaluate with precision-oriented reference metrics such as Accuracy and ROUGE-L for automatic scoring. For newly introduced open-ended generation tasks in MGAL, such as paragraph filling, we further adopt an LLM-as-a-judge method to assess quality aspects that are not fully captured by reference-based automatic metrics, following recent best practices (Zheng et al., 2023c; Liu et al., 2023b; Kim et al., 2025a). Upon comprehensive evaluation of 12 long-context LLMs on MGAL, we find that: (1) models excel on word-level tasks but struggle as evaluation shifts to coarser-grained units. and (2) large open-source and closed-source models are comparable on higher-resource languages, but closed-source systems maintain a clear advantage on lower-resource languages, with the gap more evident for smaller open-source models.

Through empirical analysis, we reveal two new key challenges: (1) under local semantic crowding where neighboring sentences contain overlapping words or repeated entities, models rely on shallow signals such as connectives (e.g., however, therefore) and repeated entities (e.g., country names, years), preferring sentences with surface overlap instead of those that fulfill the correct functional role in the paragraph (e.g., background, explanation, outcome); and (2) although outputs are often fluent and stylistically appropriate, they are weakly anchored to the input, leading to factual drift and unsupported claims. Additionally, we observe several patterns consistent with prior work. In sentence-cloze

task, error rates are highest when blanks appear early in the text and decrease toward the end, reflecting models' tendency to favor recent context due to recency-biased attention (Peysakhovich & Lerer, 2023; Hsieh et al., 2024b). We also find that models repeatedly pick earlier answer options (e.g., "A" or "B"), showing an early-position bias and reliance on option-order heuristics rather than carefully evaluating the evidence specific to each item (Pezeshkpour & Hruschka, 2024b; Zheng et al., 2023a).

Overall, our findings highlight the necessity of combining multilingual, fine-grained, and position-aware evaluation. We position MGAL as a valuable benchmark for assessment across languages and granularities, informing future long-context model development. In summary, our contributions are threefold:

- We present MGAL, the first multilingual long-context benchmark that is both granularity-aware (word, sentence, paragraph, document) and position-aware (begin, middle, end), covering all six official UN languages with contexts up to 128K tokens.

- Through a fine-grained, multilingual, and position-controlled design, MGAL decomposes evaluation along two key dimensions, i.e., linguistic granularity and evidence position, across six languages, enabling more rigorous diagnosis than existing document-level benchmarks.

- We conduct extensive evaluations of 12 long-context LLMs and use MGAL as a controlled diagnostic testbed to analyze benchmark-grounded limitations, including granularity-dependent degradation, position sensitivity, local semantic crowding, and fluency–consistency gaps.

## 2. Related work

### 2.1. Long-Context Modeling for LLMs

Long-context modeling has emerged as a key challenge for LLMs, driving innovations in both training and inference strategies. Building on rotary positional embeddings (RoPE) (Su et al., 2024), methods such as Position Interpolation (PI) (Chen et al., 2023) have demonstrated effectiveness in extending the usable context length. In parallel, sparse attention approaches further enhance scalability. For instance, LongLoRA (Chen et al., 2024) combines shifted sparse attention with LoRA, enabling models to handle up to 100k tokens at a modest computational cost. SnapKV (Li et al., 2024b) compresses the KV cache by selecting salient keys from an observation window without requiring fine-tuning, while Squeezed Attention (Hooper et al., 2025) clusters keys from the fixed portion of a prompt into centroids offline and subsequently filters the most relevant keys online, thereby accelerating inference when contexts overlap across requests. Collectively, these methods provide a practical

*Table 2.* Comparison to existing long-context benchmarks. "En", "Zh", "Es", "Fr", "Ru", and "Ar" refer to tasks in English, Chinese, Spanish, French, Russian, and Arabic,"Pos-Par." and "Pos-Doc." indicate positional evaluation at the paragraph and document level. "Gran." refers to Granularity.

| Benchmark | Max Len | En | Zh | Es | Fr | Ru | Ar | Pos-Par. | Pos-Doc. | Gran. |
|---|---|---|---|---|---|---|---|---|---|---|
| LongBench(Bai et al., 2024) | ∼10K | ✓ | ✓ | ✗ | ✗ | ✗ | ✗ | ✗ | ✓ | ✗ |
| ZeroSCROLLS (Shaham et al., 2023) | ∼10K | ✓ | ✗ | ✗ | ✗ | ✗ | ✗ | ✗ | ✓ | ✗ |
| NeedleBench (Li et al., 2024a) | ∼10K | ✓ | ✓ | ✗ | ✗ | ✗ | ✗ | ✓ | ✗ | ✗ |
| RULER (Hsieh et al., 2024a) | ∼10K | ✓ | ✗ | ✗ | ✗ | ✗ | ✗ | ✓ | ✗ | ✗ |
| LaRA (Li et al., 2025) | ∼10K | ✓ | ✗ | ✗ | ✓ | ✗ | ✗ | ✗ | ✓ | ✗ |
| M⁴LE(Kwan et al., 2024) | ∼10K | ✓ | ✓ | ✗ | ✗ | ✗ | ✗ | ✓ | ✗ | ✗ |
| LV-Eval (Yuan et al., 2024) | 4K–60K | ✓ | ✓ | ✗ | ✗ | ✗ | ✗ | ✗ | ✓ | ✗ |
| ONERULER (Kim et al., 2025b) | ∼20K | ✓ | ✓ | ✓ | ✓ | ✓ | ✗ | ✓ | ✗ | ✗ |
| **MGAL (ours)** | ∼128K | ✓ | ✓ | ✓ | ✓ | ✓ | ✓ | ✓ | ✓ | ✓ |

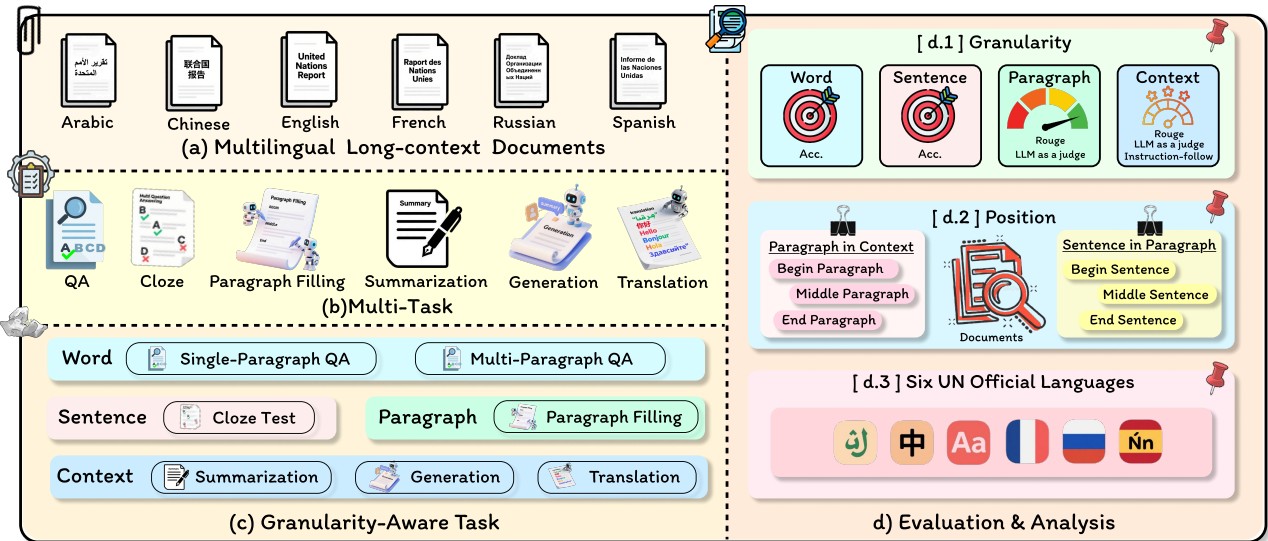

*Figure 1.* Overview of MGAL. From (a) aligned multilingual UN long-context documents, we construct (b) a multi-task benchmark covering QA, cloze, paragraph filling, summarization, and translation. These tasks are organized by (c) linguistic granularity and evaluated through (d) diagnostic analyses over granularity, position, and language.

toolbox for scaling LLMs to longer inputs while maintaining controllable cost and quality.

### 2.2. Benchmarks for Long-Context LLMs

Existing benchmarks for long-context LLMs primarily evaluate document-level comprehension and reasoning. Zero-SCROLLS (Shaham et al., 2023) targets zero-shot long-text NLU, while LongBench (Bai et al., 2023) introduces a bilingual, multi-task suite spanning single/multi-document QA and query-based summarization. To better control sequence length and task difficulty, synthetic benchmarks such as NeedleBench (Li et al., 2024a) and RULER (Hsieh et al., 2024a) have been proposed. NeedleBench adds the Ancestral Trace Challenge (ATC) for multi-step logical tracing, whereas RULER extends beyond vanilla NIAH to multi-hop tracing and aggregation. M⁴LE (Kwan et al., 2024) further expands

task and language coverage, and ONERULER (Kim et al., 2025b) increases multilingual diversity. LaRA (Li et al., 2025) complements these efforts by providing a testbed for contrasting long-context LLMs with retrieval-augmented generation(RAG) pipelines. Despite these advancements, current benchmarks offer limited insight into fine-grained discourse phenomena and lack systematic evaluation across languages and granularities. In particular, they provide little evidence on how position sensitivity and discourse consistency differ between higher- and lower-resource languages at different granularities, a gap that MGAL is specifically designed to address.We compare MGAL with existing long-context benchmarks in Table 2. MGAL supports context lengths of up to 128k tokens and covers six official languages. Moreover, it enables multi-granularity evaluation of LLMs' position sensitivity at both the paragraph and document levels.

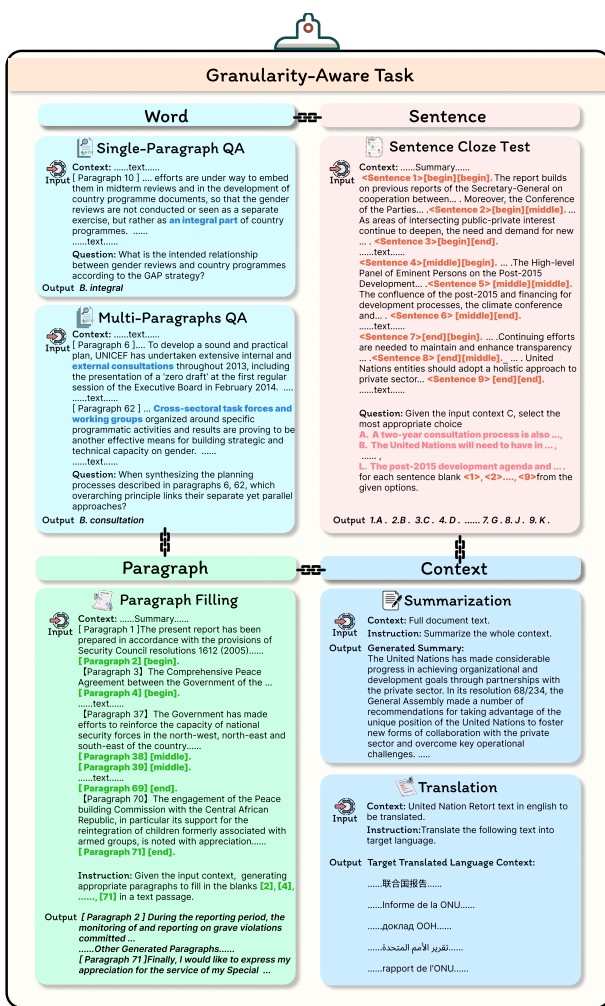

*Figure 2.* MGAL dataset instantiation at each granularity.

# 3. MGAL

We introduce **MGAL**, the first **M**ultilingual **G**ranularity-**A**ware **L**ong-context benchmark, constructed from long-form reports in the United Nations (UN) Digital Library. MGAL covers all six official UN languages (Arabic, Chinese, English, French, Russian, Spanish) with context lengths ranging from 8K to 128K tokens. It spans four linguistically coherent levels of granularity (word, sentence, paragraph, document) and stratifies instances by evidence position (begin, middle, end), enabling systematic diagnosis of multilingual long-context comprehension. We illustrate representative MGAL examples at each granularity in Figure 2 and show an overview of MGAL in Figure 1.

## 3.1. Problem Definition

In MGAL, each task is defined as taking a long-document context together with task instructions as input, and producing an output at the required granularity: word, sentence,

paragraph, or document. We list the pipeline for MGAL generation and human verification guidelines in Figure 3.

## 3.2. Dataset Construction

MGAL is built through a unified annotation and curation pipeline tailored to each granularity level. All data is constructed from a large pool of approximately 70,000 UN reports, from which we identify roughly 4,000 structurally suitable reports for benchmark construction. We sample documents across the 8K–128K token range and ensure that each task contains the same number of query–response pairs for all six languages. The resulting source documents cover diverse UN policy and governance domains, including international development, human rights, peace and security, humanitarian affairs, public health, environment, education, and economics. All generated items undergo manual verification by two trained annotators following task-specific guidelines, and only those approved by both are included. In UN reports, all paragraphs are explicitly numbered. Each document maintains the same paragraph count across all languages, and each corresponding paragraph contains the same number of sentences. Moreover, paragraphs and sentences aligned at the same positions convey equivalent semantics across language versions of the same report. All data is drawn from long-form UN reports and verified to ensure cross-lingual consistency across the six languages. Dataset statistics are summarized in table 1, with the detailed construction and human verification process provided in Appendix E.

### 3.2.1. WORD

Word-level tasks evaluate whether LLMs can identify precise words and phrases in long-context settings, probing their ability to locate and extract fine-grained evidence. We design two subtasks for evaluation: Single-paragraph Question Answering (Single-QA) and Multi-paragraph Question Answering (Multi-QA). For data construction, paragraphs are sampled from different positions in a document (begin, middle, end). We use GPT-4 (OpenAI, 2023) to generate question–answering (QA) pairs from these paragraphs, with answers annotated as word-level spans from text. All generated pairs are manually verified for correctness.

**Single-QA** The Single-QA task which constructs word-level QA pairs from individual paragraphs, with questions categorized into three types: numerical, classification, and reference. Paragraphs are sampled from different positions within the document to improve the positional diversity.

**Multi-QA** Multi-QA includes synthesis, comparison, and retrieval subtasks. Each QA pair spans two position-controlled paragraphs from the same document, requiring models to integrate evidence across both sources.

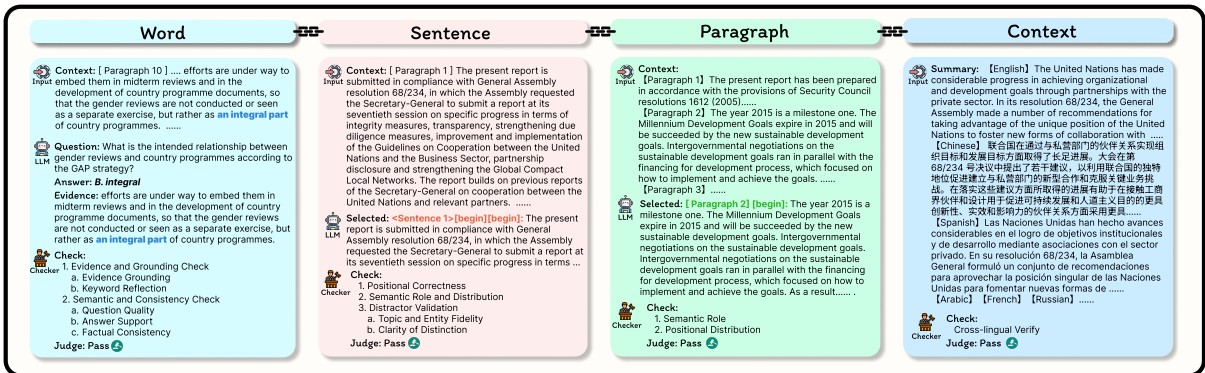

*Figure 3.* MGAL data generation and human verification pipeline. For each granularity level, candidate instances are generated from aligned UN reports and then checked by human annotators for grounding, semantic consistency, positional correctness, and cross-lingual alignment.

We provide additional details on question types for Single-QA and Multi-QA in Appendix E.1.3.

### 3.2.2. SENTENCE

**Cloze** The sentence-level task is designed to assess whether models can recognize sentence roles and maintain coherence across neighboring sentences. To this end, we introduce a novel sentence-level cloze task inspired by the 'Insert Text' question in TOEFL iBT Reading. Specifically, the task requires LLMs to recover a masked sentence using its surrounding context, ensuring both local coherence and global consistency within the document, thereby directly probing sentence-level understanding. To construct the data, we divide each document by position at both the document and paragraph levels (beginning, middle, and end). From these segments, salient sentences are extracted using GPT-4. For each instance, the selected sentence is removed and replaced with a blank. The candidate set consists of the correct sentence along with several distractors generated by GPT-4 that are semantically related but contextually inappropriate. The model must then identify the option that best restores the passage, maintaining local coherence with neighboring sentences while preserving alignment with the broader discourse.

### 3.2.3. PARAGRAPH

**Paragraph Filling** Paragraph-level tasks probe whether models can generate coherent, contextually grounded paragraphs that extend beyond the sentence scale. At this granularity, the focus is on both the understanding and generation abilities of LLMs. We design a paragraph filling task to evaluate models' capability to recover missing paragraphs using the surrounding context while maintaining the coherence of the original document. Specifically, each document is divided into position-based segments, and GPT-4 is used to identify para-

graphs with clear functional roles (e.g., topic introduction, contrast, or conclusion) whose content can be inferred from neighboring text. For each sample, the selected paragraph is removed and replaced with a blank. Unlike sentence-level cloze tasks, this requires free-form generation rather than selecting from candidates. The model is then tasked with generating a paragraph that restores local cohesion, preserves entity and event continuity, and remains aligned with the global discourse theme.

### 3.2.4. DOCUMENT

Our document-level tasks are designed to evaluate holistic document comprehension under long-context settings. Specifically, we design two tasks, summarization and translation, to assess LLMs' ability to perform comprehensive understanding and generation at the document level.

**Summarization** Summarization evaluates whether models can condense long documents into concise yet faithful summaries, testing their ability to capture salient content under extended contexts. We use human-written summaries from UN reports as reliable references. For each document, the original summary is excluded from the input, and the remaining content is given to the model, which is required to generate a faithful and informative summary under long-context conditions.

**Translation** Translation evaluates whether models can accurately preserve meaning across languages in full-document settings, probing both cross-lingual transfer and long-context comprehension. We construct six translation datasets using multilingual counterparts of the same UN documents. In each dataset, one language is fixed as the source and translated into the other five, providing a systematic and balanced setup.

# 4. Experiments

## 4.1. Experimental Setup

We evaluate long-context LLMs in a zero-shot setting on MGAL, without any fine-tuning. The evaluation details are provided in Appendix F.

### 4.1.1. MODELS

To evaluate model performance across multilingual, fine-grained long-context tasks, we benchmark 12 LLMs, covering both open-source and proprietary models, with context windows exceeding 128k tokens. The set includes GPT-5 (OpenAI, 2025), Claude Sonnet 4 (Anthropic, 2025), Gemini 2.5-Flash (Google DeepMind, 2025), Grok 4 (xAI, 2025), Doubao-Seed-1.6 (Volcengine, 2025; ByteDance Seed Team, 2025), Qwen3-235B-A22B-Instruct (Alibaba Qwen Team, 2025a), Kimi-K2-Instruct (Moonshot AI, 2025), DeepSeek V3.1 (DeepSeek, 2025), GLM-4.5 (Zhipu AI, 2025), Qwen3-30B-A3B-Instruct (Alibaba Qwen Team, 2025b), Mistral-Small-3.2-24B-Instruct (Mistral AI, 2025), and Gemma-3-27B (Google AI, 2025).

### 4.1.2. EVALUATION METRICS

We adopt task-specific metrics aligned with each level of granularity. For Word-level QA and Sentence-level Cloze, we report average accuracy to measure the proportion of predictions that match the gold labels. For Paragraph Filling and Summarization, we use ROUGE-L (Lin, 2004). For Translation, we report BLEU (Papineni et al., 2002) computed with sacreBLEU(Post, 2018). The further discussion about mertrics is provided in Appendix F.1. In addition to ROUGE-L, we employ an LLM-as-a-judge for the newly introduced open-ended paragraph-filling task in MGAL to assess quality aspect that are not fully captured by reference-based automatic metrics. Under the same evaluation guidelines used in the LLM-as-a-judge setting, we instruct our human evaluators to assess outputs in both English and Chinese. Details on prompts and implementation are provided in Appendix F.4.

## 4.2. Main Results

Table 3 and Figure 4 summarize the performance (%) of all models on MGAL (see Appendix H for complete results). On word- and sentence-level multiple-choice evaluations, GPT-5 achieves the highest average scores. At coarser granularities, Grok-4 leads on paragraph filling, while GLM-4.5 delivers the best summarization scores and Gemini-2.5-flash excel in translation. The results yield two key observations:

(1) **Fine vs. coarse performance.** Across long-context LLMs, performance is consistently strong on word-level tasks but weak at coarser granularities. Detailed analyses at each granularity are provided in Appendix G.1.

(2) **Higher- vs. lower-resource languages.** In higher-resource languages, large open-source models perform on par with closed-source systems. However, in lower-resource languages, closed-source models retain a clear advantage, with the gap more evident for smaller open-source models.

The llm-as-a-judge and human evaluation results in Appendix H.1.2 show that all models underperform on Topic Fidelity and Entity Consistency in paragraph filling. Importantly, human and LLM-as-a-judge scores show high Spearman correlation, indicating strong agreement (detailed analysis is provided in Appendix F.5). To examine whether models truly utilize the provided long context rather than relying solely on pre-training priors, we conduct a context ablation study on the Single-QA task. Detailed results are provided in Appendix G.4.

## 4.3. Positional Results at Different Granularities

Prior work shows that long-context models often pay less attention to information placed in the middle of a sequence for question answering and retrieval (Liu et al., 2023a). Similarly, as shown in Figure 5(a) and (c), accuracy in MGAL peaks when the answer lies at the boundaries and is lowest in the middle, with clear effects at the word- and paragraph-level tasks. This boundary preference suggests that models struggle to sustain effective attention over the full sequence.

By contrast, most models achieve their highest accuracy on the sentence-cloze task when the target sentence is positioned in the middle of the document as illustrated in Figure 5(b). To understand this pattern, we conduct both human and model-based analyses in Appendix G.2. Our findings suggest that, at this granularity, models tend to rely on surface overlaps and simple connectors rather than on the discourse role a sentence plays within the paragraph. Boundary sentences at the beginning and end often share framing or summary style and are easily confusable, whereas mid-paragraph sentences typically convey concrete facts tied to nearby entities and references, which better align with such surface cues. Therefore, the middle-position peak should not be interpreted as evidence of robust mid-context comprehension; instead, it likely reflects shallow cue-following and insufficient sensitivity to sentence-level roles. This insight motivates our deeper analysis of local semantic crowding and cue reliance in Section 5.1. We further disentangle document-level and paragraph-level positions for the sentence-cloze task and provide a model-wise error breakdown in Appendix G.3.

*Table 3.* Results on word, sentence, paragraph and document tasks. Open-source and proprietary models are separated with a divider, and the top-performing LLM is highlighted in bold, second best are underlined.

| Model | Word | | Sentence | Paragraph | Document | |
|---|---|---|---|---|---|---|
| | Single-QA | Multi-QA | Cloze | Filling | Summarization | Translation |
| GPT-5 | **79.79** | 74.99 | 30.62 | 14.47 | 15.47 | 35.30 |
| Claude Sonnet 4 | 72.75 | 73.76 | 21.53 | 12.81 | 22.09 | 14.96 |
| Gemini-2.5-flash | 77.01 | **75.49** | 26.00 | 15.97 | 23.89 | **36.17** |
| Grok 4 | 78.49 | 72.64 | 21.58 | **19.90** | 18.93 | 31.05 |
| Doubao-Seed-1.6 | 76.63 | 73.63 | 19.47 | 15.22 | 17.08 | 6.16 |
| Qwen3-235B-A22B | 71.84 | 73.63 | 20.29 | 14.72 | 24.27 | 9.54 |
| Kimi-K2 | 74.41 | 75.17 | 12.00 | 13.69 | 18.51 | 4.46 |
| DeepSeek-V3.1 | 75.56 | 72.50 | 17.31 | 14.54 | 26.15 | 28.52 |
| GLM-4.5 | 68.44 | 69.39 | 14.79 | 16.15 | **26.88** | 21.48 |
| Qwen3-30B-A3B | 68.86 | 71.14 | 16.54 | 13.72 | 22.74 | 10.98 |
| Mistral-Small-3.2-24B | 69.17 | 70.91 | **32.35** | 15.13 | 6.08 | 1.63 |
| Gemma-3-27B | 66.7 | 67.88 | 20.37 | 14.67 | 5.91 | 1.95 |
| **Average Performance** | 73.09 | 72.59 | 21.07 | 15.08 | 19.00 | 16.85 |

## 4.4. Ablation Studies

**Effect of Instruction Placement**  We first investigate whether model performance in the sentence-cloze task is affected by the distance between the instruction and the blank. By default, instructions and candidate options are placed at the end of the document, which means blanks appearing near the beginning are far away from the task description. To test positional sensitivity, we relocate the instruction block while keeping all blanks and candidate answers unchanged. We compare three configurations: (1) placing the instruction immediately after the beginning section (Begin), (2) placing it in the middle (Middle), and (3) the default baseline placement at the end (End). As shown in Figure 6(a), relocating the instruction improves accuracy in the nearby document region but reduces performance at more distant positions. This indicates that long-context LLMs are sensitive to instruction placement, with performance improving as the instruction moves closer to the blank.

**Effect of Option Order**  In this study, we aim to test whether LLMs in the sentence-cloze task are biased by the order of candidate options rather than by their actual content. Prior analyses suggest that models tend to repeatedly select earlier options, reflecting position bias rather than content evaluation (Pezeshkpour & Hruschka, 2024a; Zheng et al., 2023a). To probe this, we conduct an ablation where the same set of options is presented in different orders. Specifically, we test three conditions: (1) placing high-confusion distractors at the beginning (Begin), (2) placing them in the middle (Middle), and (3) placing them at the end, which corresponds to the default baseline (End). Figure 6(b) shows

that accuracy drops when distractors appear earlier (0.16 Begin, 0.21 Middle) and improves when pushed later (0.24 End). This pattern suggests that models exploit option-position heuristics, i.e., selecting answers based partly on their placement, rather than evaluating all candidates purely by semantic fit, we analyze more in Section 5.2.

## 5. More Analysis

We perform a detailed analysis based on task granularity, position, and multilingual factors, highlighting several novel challenges as well as patterns consistent with prior work.

### 5.1. New Challenges Revealed by MGAL

**Preference for Surface Cues Under Local Semantic Crowding**  At the sentence level, we observe that models tend to overweight surface cues such as explicit connectives (e.g., however, therefore), repeated entities, or overlapping lexical patterns under local semantic crowding, where neighboring sentences discuss similar topics and share entities. In such cases, models often underutilize the functional role a sentence plays within the paragraph (e.g., background, explanation, or outcome), and instead follow shallow overlaps.

For example, given an opening sentence like "The Working Group on the Universal Periodic Review, established in accordance with Human Rights Council resolution 5/1 of 18 June 2007, held its first session from 7 to 18 April 2008.", the correct next sentence should be "At its 15th meeting, the Working Group adopted the present report on Algeria." (a development sentence). However, models often prefer

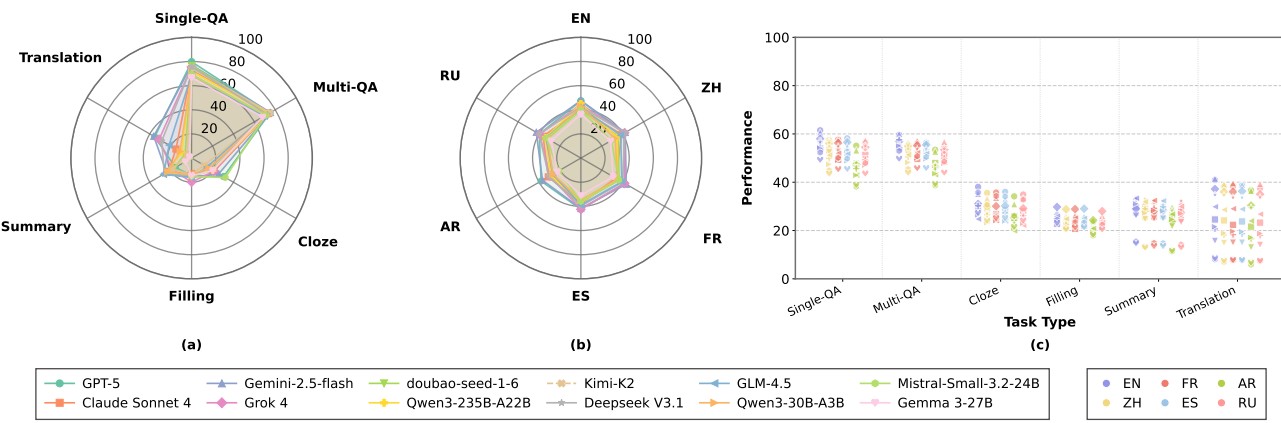

Figure 4. Performance of our evaluated models on MGAL. (a) shows LLM performance across different tasks, (b) presents multilingual performance, and (c) reports task performance across the six official UN languages. GPT-5 achieves the top average across granularity tasks, whereas Gemini-2.5-Flash excels across languages. And models achieve strong results on word-level tasks but exhibit weaker performance at coarser granularities.

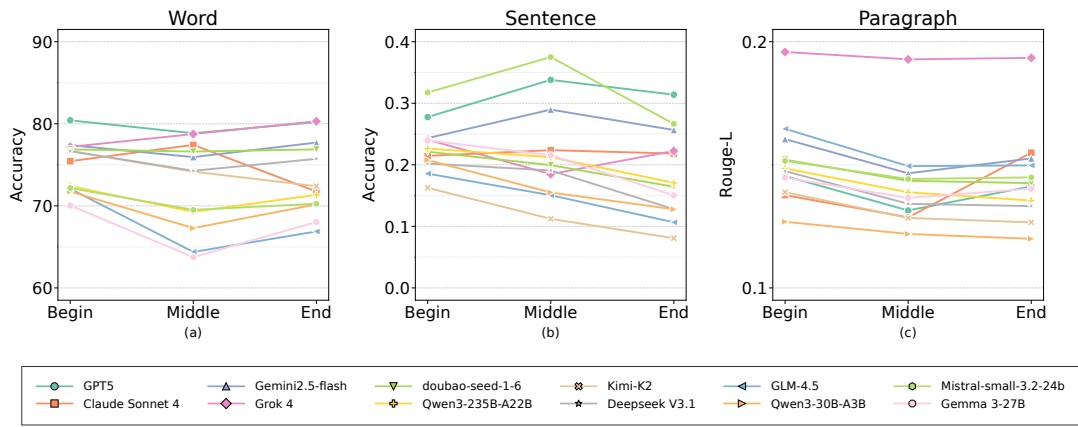

Figure 5. Position Performance on MGAL. Performance dips in the middle for word and paragraph tasks but peaks there for sentence-level tasks.

a summary-style sentence such as "The review of Algeria was held at the 11th meeting on 14 April 2008.", which appears plausible due to repeated years and entities but serves wrong discourse role, redundantly restating rather than advancing argument. This tendency shows that models rely heavily on shallow lexical overlap rather than accurately capturing sentence-level discourse roles. Similar shortcomings were documented in earlier neural models (Kim et al., 2020; Maekawa et al., 2024), and our findings suggest that long-context LLMs struggle with this challenge.

We further examine failure modes with LLM- and human-based annotations, identifying the most frequent error categories in model predictions. We focus on Cloze cases where more than 50% of models fail, using GPT-5 and Gemini-2.5-Flash to analyze the underlying error reasons. The LLMs categorize the errors and provide supporting evidence for their judgments, which are then verified by human checkers. The final results support our finding that under local seman-

tic crowding, models tend to over-rely on surface cues while under utilizing discourse-role reasoning, leading to failures in filling role-slots in patterns such as "Given $\alpha \Rightarrow \beta$", "Although $\alpha$, still $\beta$", or "$\beta$ because $\alpha$". More details are provided in Appendix G.2.

**Gap Between Fluency and Consistency in Generated Outputs** In paragraph filling and summarization tasks, model outputs often exhibit high fluency and stylistic alignment with the source document, yet they frequently neglect reliable contextual grounding. As a result, models may introduce unsupported entities or drift away from the intended explanation of the text. For example, models may overlook concrete cues in the surrounding paragraphs and confidently assert that a policy has already been implemented when the source text only outlines actionable recommendations. Similarly, they may hallucinate actors, dates, or institutions absent from the document. While the generated text reads smoothly and appears stylistically consistent, it is factually

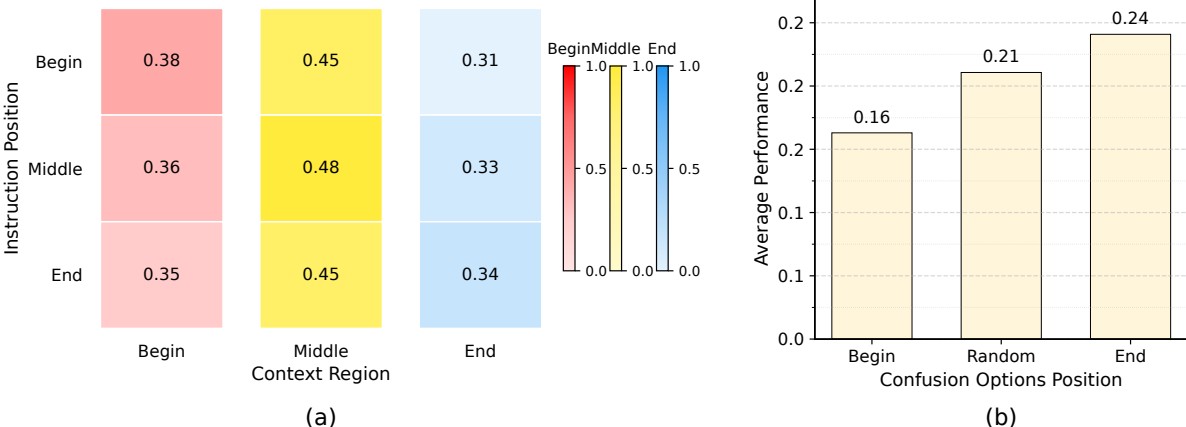

*Figure 6.* Ablation studies: (a) Accuracy rises as instruction moves nearby to the blank. (b) Accuracy drops when high-confusion distractors are positioned earlier.

inconsistent with the document, highlighting a persistent gap between fluency and consistency in long-context generation.

## 5.2. Additional Observations Consistent with Previous Studies

In addition to the novel findings, our evaluation confirms several patterns widely reported in earlier studies. Specifically, we observe well-documented tendencies such as reliance on nearby evidence and option reuse under uncertainty. These observations not only align with prior research but also complement our new insights, collectively offering a more comprehensive characterization of long-context model behavior.

**Reliance on Nearby Evidence**  In the sentence-cloze task, blank omissions peak when the blank appears near the beginning of the document and decline toward the end (Appendix G.1). This aligns with prior work showing that long-context models underutilize distant evidence due to recency-weighted attention, and that repositioning salient segments closer to the decoding point can mitigate this limitation (Peysakhovich & Lerer, 2023). Related studies on positional calibration and controlled placement also demonstrate that model performance is highly sensitive to the relative distance between evidence and the decision anchor (i.e., the instruction and candidate options) (Hsieh et al., 2024b; Xu et al., 2024). In our setup, the anchor is appended after the document, effectively anchoring the decision at the end of the input. As a result, blanks at the beginning correspond to the greatest anchor–evidence distance, leading to more omissions. Following the ablation in Section 4.4, we relocate instructions, and find that it reduces omissions at the corresponding position, which further suggests our explanation.

**Reuse of Options Under Uncertainty**  When uncertain, models tend to repeatedly select the same options, disproportionately favoring those in earlier positions, such as A and B over C and D in Appendix G.1. This behavior reflects an early-position bias, suggesting that models rely on option-order heuristics and frequency priors rather than grounding their choices in item-specific evidence (Pezeshkpour & Hruschka, 2024b; Zheng et al., 2023a). The option order shuffling ablation in Section 4.4 further supports that placing confusion distractors earlier degrades models' accuracy, while placing them later improves it.

## 6. Conclusion

We introduced **MGAL**, the first multilingual benchmark for evaluating long-context LLMs across multiple levels of granularity and controlled positional settings. Built from UN reports spanning 8K–128K tokens in six official languages, MGAL covers four linguistic units (word, sentence, paragraph, document) and systematically varies evidence positions, enabling multilingual fine-grained and position-aware assessment. Through an extensive evaluation of 12 LLMs, we find that while models perform relatively well on fine-grained QA, their performance is weak on coarser tasks. We further identify two new challenges specific to long contexts: *local semantic crowding*, where models over-rely on surface cues instead of recognizing discourse roles, and a *fluency–consistency gap*, where generated outputs remain stylistically fluent yet factually misaligned with the source. In addition, MGAL confirms previously observed weaknesses such as recency bias and option-order heuristics. Overall, MGAL highlights the limitations of current LLMs in multilingual long-context comprehension and establishes a rigorous testbed for guiding future work on training objectives and evaluation methodologies that emphasize robust, fine-grained, and position-sensitive understanding.

## Impact Statement

Our benchmark is constructed from publicly available reports from the United Nations (UN) Digital Library, used with respect to their public licenses and without altering their substantive meaning; we credit the UN as the original rightsholder. We screened all instances to avoid including sensitive personal data and directed human annotators to exclude content with offensive language or social biases. While the potential for encountering sensitive content from the original sources persists, this risk is mitigated as the benchmark's primary focus is on evaluating the long-context capabilities of LLMs, not their social biases. To further protect privacy, released artifacts contain only minimum text necessary for evaluation, with full documents remaining at their original sources. Furthermore, our methodology considers environmental impact by evaluating existing models without additional pre-training and reporting settings. Finally, we will comply with any takedown or correction requests from rights holders or affected parties and will promptly update dataset documentation if legal interpretations change.

## Acknowledgement

This work is supported by the Youth S&&T Talent Support Programme of GDSTA (SKXRC2025462), NSFC (Category C) fund code 62506149 and Lingnan University StartUp Grant fund code: 103684.

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

## A. Reproducibility Statement

We ensure reproducibility in three aspects:

(1) Dataset. All source documents are drawn from the publicly accessible UN Digital Library. The data curation and annotation pipeline, including sampling strategies for each granularity, is fully documented, and the processed benchmark will be released upon publication with task splits.

(2) Evaluation. We provide the exact prompts, scoring scripts, and task definitions for all evaluations.

(3) Models and code. For open-source models, we will release configuration files and inference scripts upon publication; for API-based models, we document request formats and parameters. All preprocessing, annotation, and evaluation code will be released upon publication.

## B. The Use of Large Language Models

To enhance the clarity and readability of this manuscript, we utilized Large Language Models as assistive tools. First, we employed them for language refinement, including grammatical correction, stylistic improvements, and the rephrasing of complex sentences. Second, we leveraged LLMs to support our data construction pipeline. Specifically, LLMs assisted in initial annotation and cleaning of each granularity data set derived from UN documents; these data subsequently underwent final validation by human checkers. The authors maintained full editorial control throughout this process. All substantive contributions, including ideas, methodology, analyses, and conclusions, are exclusive work.

## C. Limitations

Although MGAL broadens the evaluation scope for fine-grained long-context understanding, several limitations remain. First, standard automatic, reference-based metrics (e.g., ROUGE-L for summarization; token-level accuracy for QA; BLEU for translation) are coarse proxies for human judgment and are sensitive to surface form, paraphrase, and length, which can underestimate or mischaracterize quality (Novikova et al., 2017; Reiter, 2018; Lin, 2004). Second, using LLM-as-a-judge improves semantic sensitivity but incurs nontrivial runtime cost and exhibits known biases (e.g., position and verbosity), requiring careful prompt design, calibration, and robustness checks (Zheng et al., 2023b). Third, while our goal is to evaluate long-context modeling independently of instruction following, real-world task formulations inevitably intertwine the two, so measured performance may partly reflect instruction adherence rather than pure context-modeling ability.

## D. UN Language Scope and Relative Resource Imbalance

MGAL covers the six official languages of the United Nations: Arabic, Chinese, English, French, Russian, and Spanish. All six languages are high-resource languages by conventional standards, with large speaker populations: English (1.45B), Chinese/Mandarin (1.13B), Spanish (559M), French (310M), Arabic (274M), and Russian (255M). Our analysis on MGAL concerns relative resource imbalance within this high-resource subset, highlighting cross-language performance differences without conflating them with genuinely low-resource languages.

## E. Dataset Construction Pipeline

This section describes the dataset construction pipeline for MGAL. We source documents from the United Nations Digital Library and apply manual annotations without altering substantive content.

MGAL is built through a unified annotation and curation pipeline tailored to each granularity level. All data is constructed from a large pool of approximately 70,000 UN reports, from which we identify roughly 4,000 structurally suitable reports for benchmark construction. We sample documents across the 8K–128K token range and ensure that each task contains the same number of query–response pairs for all six languages. The resulting source documents cover diverse UN policy and governance domains, including international development, human rights, peace and security, humanitarian affairs, public health, environment, education, and economics.

In UN reports, all paragraphs are explicitly numbered. Each document maintains the same paragraph count across all languages, and each corresponding paragraph contains the same number of sentences. Moreover, paragraphs and sentences aligned at the same positions convey equivalent semantics across language versions of the same report.

Each generated instance is checked by two trained annotators for correctness, contextual grounding, positional validity, and cross-lingual consistency, and only instances approved by both annotators are retained. Across tasks, the accepted/generated counts are 420/582 for Single-QA, 420/604 for Multi-QA, 420/800 for Cloze, 420/500 for Paragraph Filling, and 420/800 for Summarization. Rejected QA items mainly involve semantic ambiguity, shortcut cues that allow answering without true contextual grounding, or conflation of distinct paragraph meanings. Rejected cloze and paragraph-filling items are primarily due to unbalanced position or role selection, while rejected summarization items often arise from formatting inconsistencies in UN reports that cause extraction or cross-lingual alignment errors.

## E.1. Word QA

For the referenced single- or multi-paragraph inputs, the LLM is prompted to generate a question, an answer, and the corresponding evidence from each specified paragraph according to the category-specific instructions. The human checkers evaluate each generated question–answer pair following a two-stage guideline. First, they verify whether the provided evidence is grounded in the input paragraph, rather than being hallucinated by the LLM. Second, the checkers assess the quality of the question–answer pairs. They begin by examining the semantic correctness and clarity of the generated question, ensuring that it is unambiguous and aligned with the predefined categories. They then evaluate whether the answer is supported by the evidence within the paragraph and whether the question–answer pair is consistent and factually correct.

### E.1.1. SINGLE QA

We partition each document's main body into three position-indexed regions: begin, middle, and end. From each region, we robustly extract complete paragraphs. One paragraph is uniformly sampled per region. For each sampled paragraph, we prompt an LLM to generate a single question that includes cited paragraph-bounded evidence, and requires a word or phrase answer sourced from the original paragraph. We categorize three generated question templates as: (1) Numerical for exact counts and quantitative mentions; (2) Classification for topic, sentiment, or functional category; and (3) Reference for pronoun or coreference resolution with local inference.

Each question is formatted as a four-choice question with a single-answer option. The options distractors are constructed to be semantically plausible under partial reading yet inconsistent with the anchored evidence. At evaluation time, models are given the full document and the question that instructs them to answer.

```
You need to analyze the following paragraph and create a question that tests the ability to find specific numerical information.

Paragraph {paragraph_id}: {source_text}

Step-by-step process:
    1. Identify all numerical data, statistics, measurements, or quantitative information in paragraph {paragraph_id}
    2. Select the most significant or central numerical value
    3. Create a question that requires locating this specific number
    4. Ensure the question cannot be answered through general knowledge
    5. Extract 1-2 grounded evidence summaries that support the numerical answer

Requirements: ......

CONCRETE EXAMPLES:  Few shot examples......

Output format:
  "question": "Based on paragraph {paragraph_id}, what [numerical aspect] is mentioned/reported/indicated?",
  "correct_answer": "number",
  "evidence":
    {{"paragraph_id": {paragraph_id}, "evidence": "concise grounded summary supporting the numerical answer"}}
```

*Figure 7.* Prompt template for the Numerical type question generation.

You need to analyze the following paragraph and create a question that tests deep understanding and reference resolution.

Paragraph {paragraph_id}: {source_text}

Step-by-step process:
    1. Identify pronouns, implicit relationships, or logical connections in paragraph {paragraph_id}
    2. Determine what interpretation or inference is needed to understand these relationships
    3. Create a question that tests this understanding and reference resolution
    4. Ensure the answer requires non-trivial understanding, not just surface reading
    5. Extract 1-2 grounded evidence summaries that support the interpretation and inference

Requirements: ......

CONCRETE EXAMPLES: Few shot examples......

Output format:
  "question": "Based on paragraph {paragraph_id}, what does [pronoun/concept] refer to or what can be inferred about [logical relationship]?",
  "correct_answer": "inferred_concept",
  "evidence": [
    {{"paragraph_id": {paragraph_id}, "evidence": "concise grounded summary supporting the interpretation/reference resolution"}}

*Figure 8.* Prompt template for the Reference type question generation.

You need to analyze the following paragraph and create a question that tests classification or categorization abilities.

Paragraph {paragraph_id}: {source_text}

Step-by-step process:
    1. Analyze the overall tone, theme, and characteristics of paragraph {paragraph_id}
    2. Identify what category, sentiment, or classification best describes this paragraph
    3. Create a question that tests recognition of this classification
    4. Ensure the classification is specific to this paragraph's content
    5. Extract 1-2 grounded evidence summaries that support the classification

Requirements: ......

CONCRETE EXAMPLES: Few shot examples......

Output format:
  "question": "Based on paragraph {paragraph_id}, what [category/theme/sentiment/approach] does this paragraph represent?",
  "correct_answer": "classification",
  "evidence": [
    {{"paragraph_id": {paragraph_id}, "evidence": "concise grounded summary supporting the classification"}}

*Figure 9.* Prompt template for the Classification type question generation.

## E.1.2. MULTI QA

We adopt the same position segmentation as Single-QA, partitioning each document's main body into begin, middle, and end regions. We sample the anchor paragraph pairs for one question-answer pair to test the models' ability to integrate contextual evidence. The sampled paragraph pairs' positions are both within regions (eg, Begin and Begin) and across regions (eg, Begin and Middle). Each question references exactly two paragraphs.

Before composing the question, the LLM produces the selected paragraph evidence summaries for each of the two selected paragraphs. These summaries anchor subsequent question wording and option construction. We categorize three cross-paragraph question templates: (1) Comparison that contrasts methods, perspectives, or claims across the two paragraphs; (2) Retrieval that locates where a specific fact resides with fixed options: first paragraph, second paragraph, both, or neither; and (3) Synthesis that derives a concept or conclusion that emerges only when the two paragraphs are considered together.

At evaluation time, models are given the full document, and the instructions for the model to answer with respect to those paragraphs questions.

```
You need to analyze the following two paragraphs and create a high-comprehension question about their combined meaning or
conclusion.

Content from paragraphs {para1_id} and {para2_id}: {paragraph_text}

Step-by-step process:
     1. Identify the main concept or theme in each paragraph
     2. Determine how they relate to each other or what they collectively demonstrate
     3. Find the synthesized conclusion that emerges from both paragraphs
     4. Create a question that tests this integrative understanding
     5. Extract 2-3 grounded evidence summaries with paragraph ids that best support the synthesized conclusion

Requirements: ......

CONCRETE EXAMPLES:  Few shot examples......

Output format:
  "question": "Based on {para_desc} together, what [conclusion/concept/pattern] emerges?",
  "correct_answer": "synthesis_concept",
  "evidence":
          {{"paragraph_id": {paragraph_ids[0] if len(paragraph_ids)>0 else 1}, "evidence": "concise grounded summary supporting
the conclusion"}},
          {{"paragraph_id": {paragraph_ids[1] if len(paragraph_ids)>1 else 2}, "evidence": "concise grounded summary supporting
the conclusion"}}
```

*Figure 10.* Prompt template for the Synthesis type question generation.

You need to create a COMPREHENSION question that tests the ability to understand information and identify which paragraph contains the answer to a specific question.You are provided with Paragraph {para1_id} and Paragraph {para2_id}. The question MUST explicitly mention both paragraph numbers.

Content from paragraphs {para1_id} and {para2_id}: {paragraph_text}

Step-by-step process:
  1. Analyze both paragraphs thoroughly
  2. Decide which answer location type to use (A, B, C, or D)
  3. Based on the type:
   - For Type A/B: Find unique information in that specific paragraph
   - For Type C: Identify something plausible but NOT mentioned in either paragraph
   - For Type D: Find information that spans or requires synthesis from both paragraphs
  4. Create a question that CONTAINS the answer/statement and asks for its location
  5. The question must explicitly reference both paragraph numbers

ANSWER LOCATION TYPES (choose one):
  - Type A: Information found ONLY in Paragraph {para1_id}
  - Type B: Information found ONLY in Paragraph {para2_id}
  - Type C: Information found in NEITHER paragraph (ask about something NOT mentioned)
  - Type D: Information requiring BOTH paragraphs to answer completely

Requirements: ......

CONCRETE EXAMPLES:  Few shot examples......

Output format:
 "question": "Between Paragraph {para1_id} and Paragraph {para2_id}, [rest of question with specific fact/answer]",
 "correct_answer": "Paragraph {para1_id}|Paragraph {para2_id}|Neither|Both",
 "evidence":
   {{"paragraph_id": {para1_id}, "evidence": "concise summary showing relevant content (or lack thereof)"}},
   {{"paragraph_id": {para2_id}, "evidence": "concise summary showing relevant content (or lack thereof)"}}

*Figure 11.* Prompt template for the Retrieval type question generation.

You need to analyze the following two paragraphs and create a high-comprehension question about their differences or contrasts.

Content from paragraphs {para1_id} and {para2_id}: {paragraph_text}

Step-by-step process:
  1. Analyze the main approach, method, or perspective in each paragraph
  2. Identify the key difference or contrast between them
  3. Create a question that tests recognition of this difference
  4. Ensure the difference is significant and not superficial
  5. Extract 2-3 grounded evidence summaries with paragraph ids that best support the identified difference

Requirements: ......

CONCRETE EXAMPLES:  Few shot examples......

Output format:
 "question": "Based on {para_desc}, what is the main difference in their [approach/method/perspective]?",
 "correct_answer": "difference_concept",
 "evidence":
   {{"paragraph_id": {paragraph_ids[0] if len(paragraph_ids)>0 else 1}, "evidence": "concise grounded summary from this paragraph supporting the difference"}},
   {{"paragraph_id": {paragraph_ids[1] if len(paragraph_ids)>1 else 2}, "evidence": "concise grounded summary from this paragraph supporting the difference"}}

*Figure 12.* Prompt template for the Comparision type question generation.

E.1.3. INSTANTIATIONS OF DIFFERENT QA TASK SAMPLE

| Task Name | Single-QA |
|---|---|
| Type | Numerical |
| Paragraph_id | 84 |

| | |
|---|---|
| Source_text: | Finally, as changing gender norms and stereotypes takes considerable time, UNDP needs longer-term, multisectoral interventions with predictable financing. To significantly contribute to the achievement of the 2030 Agenda, UNDP must work more closely with partner agencies across the United Nations system to advocate for new and sustained funding models. |
| Question: | Which year in paragraph 84 represents the culmination of UNDP's efforts to address changing gender norms? |
| Options: | A.2025 B.2030 C. 2040 D. 2020 |

*Figure 13.* Numerical Type Sample Question.

| Task Name | Single-QA |
|---|---|
| Type | Classification |
| Paragraph_id | 80 |

| | |
|---|---|
| Source_text: | The Special Rapporteur recommends that the Government of Eritrea: Put an immediate end to human rights violations documented by the Special Rapporteur and the commission of inquiry on human rights in Eritrea, including the ongoing violations highlighted in the present report; ..… Investigate the allegations of human rights and humanitarian law violations by Eritrean armed forces in the context of the conflict in Ethiopia since November 2020 and take measures to bring perpetrators to justice; (l) Refrain from subjecting Indigenous communities to discriminatory practices, including arbitrary arrests, and respect and protect their traditional ways of life and means of livelihood; |
| Question: | What conclusion can be drawn from paragraph 80's emphasis on actions for improvement? |
| Options: | A. observations B. conclusions C. suggestions D. recommendations |

*Figure 14.* Classification Type Sample Question.

| Task Name | Single-QA |
|---|---|
| Type | Reference |
| Paragraph_id | 22 |

| Source_text: | The IEAS continued to support work led by management on the Anti-Fraud program at UN-Women. Among others, it prepared a lessons-learned memorandum on red-flags/potential fraud risks related to managing implementing partners. It also initiated a lessons learned/integrity review of vehicle management, facilitated reporting on potential allegations to OIOS, and continued to support OIOS on its reports and referrals. IAS also assisted management in preparing the fraud assessment and fraud prevention training. The ACO is pleased to see the increased focus on anti-fraud awareness raising and training to improve the low-level maturity rating of this capacity as reported in the 2021 ACO report to the EB. |
|---|---|
| Question: | What role does paragraph 22 suggest the ACO takes in relation to the Anti-Fraud program? |
| Options: | A. directive B. adversarial C. indifferent D. supportive |

*Figure 15.* Reference Type Sample Question.

| Task Name | Multi-QA |
|---|---|
| Type | Comparison |
| Paragraph_id | 88,92 |

| Source_text: | 88. Given continued grave violations against children in Somalia, I call upon all responsible United Nations bodies to ensure that the protection of children is addressed as a priority in the ongoing peace process child protection advisers should be incorporated in the United Nations Political Office for Somalia, and in any future deployment of a United Nations peacekeeping operation, to serve as interlocutors with child protection actors. 92. My Special Representative for Children and Armed Conflict is requested to undertake a mission to Somalia in the near future to assess first-hand the situation for children and the implementation of the recommendations in my reports and those of the Security Council Working Group on Children and Armed Conflict." |
|---|---|
| Question: | Which cross-paragraph concept in 88, 92 illustrates the necessary steps towards enhancing child protection? |
| Options: | A. prevention B. implementation C. negotiation D. observation |

*Figure 16.* Comparision Type Sample Question.

| Task Name | Multi-QA |
|---|---|
| Type | Synthesis |
| Paragraph_id | 16,18 |
| Source_text: | 16. At its second regular session of 2015, CEB endorsed the global initiative on\ndecent jobs for youth. Prepared through an inter-agency consultative process under the leadership of the International Labour Organization … on the promotion of sustained, inclusive and sustainable economic growth, full and productive employment and decent work for all. 18. One initiative that gained particular momentum in 2015 was the United Nations system data catalogue project, the aim of which is to maximize the benefits of making United Nations system data open and accessible to the public and other key stakeholders,…As of the end of 2015, the catalogue comprised nearly 4,000 data sets. An initial public launch of the data catalogue in 2016 is foreseen. III. Promoting system-wide preparation for and follow-up to United Nations conferences and summits. |
| Question: | Which synthesis element across paragraphs 16, 18 demonstrates their interconnected approach to achieving goals? |
| Options: | A. innovation B. independence C. collaboration D. competition |

*Figure 17.* Synthesis Type Sample Question.

| Task Name | Multi-QA |
|---|---|
| Type | Retrieval |
| Paragraph_id | 4,28 |
| Source_text: | 4.In the same resolution, the General Assembly welcomed efforts to increase theeffectiveness, accountability and credibility of the United Nations system, includingby reducing administrative and procedural burdens. …The Assembly made additional requests relating to a coordinatedapproach to multilingualism, the mainstreaming of support for South-Southcooperation and the continuation of dialogue between CEB and Member States. 28. At the session, the collective engagement of the United Nations system was\ncoordinated and streamlined with a view to making the climate-related knowledge …… Enhancing the effectiveness, efficiency, coherence and impact of United Nations operational activities for development. |
| Question: | Comparing Paragraph 4 and Paragraph 28, and noting that Paragraph 28 focuses on UN system climate coordination, in which paragraph (Paragraph 4 or Paragraph 28) is the explicit statement found that the General Assembly welcomed efforts to increase effectiveness and accountability? |
| Options: | A. Paragraph 4 B. Paragraph 28 C. Neither D. Both paragraphs |

*Figure 18.* Retrieval Type Sample Question.

## E.2. Cloze

We partition each document's main body into three position-indexed regions, and identify paragraphs and select key sentences stratified by paragraph-level position within each region by LLM. For documents of 8K–16K tokens, we select nine target sentences to cover both document-level and paragraph-level positions; for documents over 16K tokens, we select twelve target sentences.

For each selected sentence, we use an LLM to generate confusion sentences that are locally topical yet inconsistent with the exact entailment required at the blanked location. We then assemble a fixed-size option set: 10 choices (9 gold + 1 confusion) for 8K–16K-token documents and 14 choices (12 gold + 2 confusion) for documents over 16K tokens. For chapter-level inputs divided into beginning, middle, and ending sections at both the paragraph and document levels, the LLM is prompted to identify key sentences (e.g., core or turning points) at the specified positions. Based on these selected sentences, the LLM then generates distractors that alter certain details while remaining topically consistent with the original content.

To verify quality, human evaluators first visualize each LLM-selected sentence in the original UN PDF. They check the positional correctness of the extracted sentences using both programmatic and manual methods. Since UN reports provide explicit paragraph numbering and sentences are segmented by periods, the evaluators can automatically confirm the sentence index and its exact location within the document. After confirming the position, the evaluators examine the semantic role of each selected sentence by inspecting the visualized sentences in the PDF to determine whether it functions as a core, transitional, or summary statement. They also verify that the selected sentences are evenly distributed across positional categories within the document to avoid selected bias.

For distractor validation, the evaluators assess topic fidelity and entity consistency to ensure that each distractor remains aligned with the referenced sentence while introducing incorrect or altered details relative to the ground truth. They then compare every distractor with all correct answer options in the cloze to guarantee a clear and unambiguous distinction between the correct choice and the distractors.

In the evaluation, the input is the full document with the target sentence replaced by a blank marker at its original location, the corresponding set of options, and an instruction that prompts LLM to select the sentence that best aligns with the local context.

## E.3. Paragraph Filling

We partition each document into the Begin, Middle, and End regions. Within each region, we use LLM to extract key paragraphs. For 8K–16K-token documents, we select two target paragraphs per region; for documents over 16K tokens, we select three per region, ensuring balanced coverage across regions.

Human evaluators then examine each LLM-selected paragraph by visualizing it directly in the original UN PDF. Using this visualization, the evaluators verify the semantic role of the paragraph, assessing whether it functions as a core, transitional, or summary element, and additionally confirm that the selected paragraphs are evenly distributed across positions within each document to avoid positional bias.

At evaluation time, we remove the paragraph and insert a blank marker at its original location. The model receives the document replaced by blank marker and an instruction to reconstruct the missing paragraphs using surrounding context.

## E.4. Context

For the summarization task, we programmatically extract the original summary from each document and perform cross-lingual verification by matching it against the corresponding summary section in the UN PDF across all six language versions.

# F. Experimental Setup

This section describes the experimental setup for MGAL, covering output controls, context management, tool-use constraints for agentic models, the judging protocol for generative tasks, and the evaluation prompts. We locally deploy Qwen3-30B-A3B-Instruct (Alibaba Qwen Team, 2025b), Mistral-Small-3.2-24B-Instruct (Mistral AI, 2025), and Gemma-3-27B (Google AI, 2025), while the remaining larger models are accessed via API.

## F.1. Interpreting Metrics Across Granularities

Tasks at different granularities in MGAL adopt different evaluation metrics: accuracy for word- and sentence-level QA, ROUGE-L for paragraph filling, and ROUGE-L or BLEU for document-level summarization and translation. Since these metrics are not directly comparable in absolute value, we emphasize relative performance trends rather than raw scores when analyzing results across granularities.

Specifically, we interpret higher accuracy on word- and sentence-level QA as evidence that fine-grained comprehension is easier for current LLMs, while substantially lower ROUGE-L scores on paragraph filling and summarization indicate persistent difficulty in generating coherent and factually grounded outputs at larger units. This rationale allows us to compare trends across tasks without conflating metric scales, and ensures that our conclusions about fine- versus coarse-grained performance reflect relative difficulty rather than absolute score magnitudes.

## F.2. Maximum Output Length Controls

To prevent non-stop generation, we cap the model's maximum output length per task/dataset. For classification-style tasks (e.g., Cloze, Single-QA, Multi-QA), we enforce single-token or single-word answers via explicit instructions and post-hoc normalization.

## F.3. Tool-use Control for Agent-based LLMs

For models with agent capabilities (e.g., tool calls, browsing, retrieval), we disable external tools and explicitly instruct the model to rely solely on the provided text. This prevents access to outside knowledge sources or caches and yields a fair cross-model comparison under identical evidence exposure.

## F.4. LLM-as-a-Judge for Paragraph-filling

For paragraph filling, we adopt LLM-as-a-judge for evaluation. The judge is given the previous and next paragraphs of the paragraphs that need to be generated, the gold reference, and the system output, and is instructed to ground every decision in the provided text rather than using external knowledge or chain-of-thought rationales.

Motivated by (Kim et al., 2025a), we perform scoring on five dimensions: Topic Fidelity, Local Coherence, Entity Consistency, Instruction Following, and Format Compliance. Each dimension is scored as an integer from 0 to 20, and the overall score is the sum 0 to 100. We prompt LLMs' judgment should be justified with brief, text-grounded reasons that cite evidence from the original document and include a short quote from a previous paragraph, ground truth, and the next paragraph.

In Topic Fidelity, the judge verifies that the generation preserves the central topic and key claims of the reference without improperly narrowing or expanding its scope, and without introducing unsupported assertions. Local Coherence assesses logical and temporal continuity between the generation and its neighbors, including transitions, pronoun and tense alignment, and causal or contrastive links. Entity Consistency checks that subjects, events, times, locations, organizations, numbers, and coreference match the reference and the constraints implied by the surrounding context, with no invented attributes. Instruction Following evaluates adherence to the task constraints (e.g., style, perspective, length, prohibited content) as specified to the judge. Format Compliance evaluates whether the output conforms exactly to the required schema or template, including structure, headings, bulleting, tags, field ordering, and length limits.

We use advanced closed-source models GPT-5, Gemini-2.5, and Grok-4 as judges and report the average evaluation score. For the LLM-as-a-judge setting in the paragraph-filling task, the judges evaluate each generated paragraph using the ground-truth paragraph and its surrounding context. The detailed inputs and prompts are provided in Appendix F.4. Each LLM judge outputs both a score and explanatory evidence. Human evaluators then verify the score and the accompanying evidence against the generated paragraph and the ground truth to ensure that the LLM's judgment is faithful, well-grounded,

and free from hallucinated evidence.

We report the mean score using GPT-5, Gemini-2.5-flash and Grok 4 as a judge and note robustness checks by humans. The exact prompt and field-level instructions used by the judge are shown in the Figure 19.

```
"You are an impartial academic evaluator. Your task is to judge the GENERATED paragraph ("gen") against the REFERENCE
paragraph ("ref") with access to its immediate context ("prev" and "next"). Rely ONLY on the provided text (no external
knowledge). Do not rewrite anything; just evaluate.

prev: {prev}
ref: {ref}
next: {next}
gen: {gen}

Score each dimension as an INTEGER from 0 to 20. Give clear, text-grounded reasons with quotes (≤30 words) from prev/
ref/next whenever deducting points.

1) Topic Fidelity (0–20)
  - Measures whether gen preserves the central topic and key claims of ref without narrowing/expanding scope improperly
or introducing unsupported assertions.

2) Local Coherence w.r.t. Context (0–20)
  - Checks logical/temporal continuity and discourse flow between gen and its neighbors (prev, next): transitions, pronoun/
tense alignment, causal/contrast links.

3) Entity Consistency (0–20)
  - Verifies that subjects, events, time, locations, organizations, numbers, and coreference in gen match ref (and constraints
implied by prev/next). No invented attributes.

4) Instruction Following (0–20)
  - Evaluate gen against the task instructions and constraints below. Penalize any ignored or violated requirement (e.g.,
style, perspective, length, prohibited content).
  - Task Instructions: {task_instructions}

5) Format Compliance (0–20)
  - Check whether gen matches the required output format/schema/template exactly (structure, headings, bulleting, tags,
fields, ordering, length constraints, etc.).
  - Expected Format Rules: {expected_format_rules}

OUTPUT:"""
```

*Figure 19.* The prompt for LLM-as-a-judge.

### F.5. Human Evaluation for Paragraph-filling

Under the same evaluation guidelines used in the LLM-as-a-judge setting, we instructed our human evaluators to assess outputs in both English and Chinese.

We calculated the relationships between different indicators in the Average row using the Pearson Correlation Coefficient, comparing the score sequences of all indicators across the LLM-as-a-judge and human evaluation tables. From the average scores of all models, there is a strong positive correlation (r = 0.871) between the LLM-as-a-judge indicators and the corresponding human assessment indicators. Consistently, both evaluations reveal the same fluency–consistency gap that models achieve high fluency scores but exhibit low consistency.

The complete human evaluation results are provided in the Appendix H.1.2.

### F.6. Human Involvement and Quality Assurance

Human involvement played a central role in the annotation verification and quality assurance process. All human annotators were recruited internally and enumerated through authorship. Specifically, we employed eight human checkers, each of whom independently participated in the full annotation verification pipeline. No crowdsourcing platforms or external contractors were used.

Each annotator contributed approximately 90 human-hours to quality assurance, corresponding to a workload of 3 hours per day over a 30-day period. In total, this resulted in 720 human-hours dedicated exclusively to manual checking, verification, and consistency validation.

Human review was applied at multiple stages of the pipeline. Annotators were responsible for validating extracted content, verifying alignment with source documents, and identifying annotation errors or inconsistencies. All instances were manually reviewed rather than sampled. When potential issues were identified, annotations were either corrected or rejected following discussion among the annotators.

Disagreements among annotators occurred occasionally, primarily in ambiguous cases involving boundary decisions or semantic interpretation. Such disagreements were resolved through discussion and consensus among the eight contributors. Questions or annotations that failed to meet quality standards after review were rejected rather than force-labeled, ensuring that only high-confidence instances were retained in the final dataset.

## F.7. Evaluation Prompts

We standardize task prompts across languages and granularities; all prompts are multilingual with explicit answer-format constraints.

> You are tasked with answering a multiple-choice reading comprehension question based on the provided document. You need to carefully read and understand the document, then select the most appropriate answer from the given options.
>
> DOCUMENT:
> {document_body}
>
> QUESTION:
> {question}
>
> OPTIONS:
> {opts_joined}
>
> INSTRUCTIONS:
> 1. Read the document thoroughly and understand its content
> 2. Analyze the question carefully to understand what is being asked
> 3. Review all four options (A, B, C, D) and evaluate which one best answers the question based on the document
> 4. Consider the context, details, and specific information mentioned in the document
> 5. Select the option that is most accurate and directly supported by the document content
> 6. Provide ONLY the single capital letter (A, B, C, or D) as your answer
> 7. Do not include any explanation, reasoning, or additional text
>
> OUTPUT FORMAT:
> [Single letter: A, B, C, or D]
>
> ANSWER:"""

*Figure 20.* An example prompt for the QA task.

> You are tasked with filling in the blanks in a text passage. The text contains numbered blanks marked as <1>, <2>, <3>, etc. You need to select the most appropriate choice for each blank from the given options.
>
> TEXT WITH BLANKS:
> {original_text}
>
> ANSWER CHOICES:
> {choices_formatted}
>
> INSTRUCTIONS:
> 1. Read the text carefully and understand the context
> 2. For each numbered blank <1>, <2>, <3>, etc., select the most appropriate choice from the given options
> 3. Fill the blanks in sequential order (<1> first, then <2>, then <3>, etc.)
> 4. Do not use any web search tools or output thought chains, only answer based on the knowledge of the model itself.
> 5. Output ONLY the answers in the format: number:letter, separated by commas. Do not output the thinking chains.
> 6. Example output format: 1:A, 2:B, 3:C
>
> OUTPUT:"""

*Figure 21.* An example prompt for the Cloze task.

> You are tasked with generating appropriate paragraphs to fill in the blanks in a text passage. The text contains numbered blanks marked as [1], [2], [3], etc. You need to generate coherent and contextually appropriate paragraphs for each blank based on the surrounding context.
>
> TEXT WITH BLANKS:
> {original_text}
>
> BLANKS TO FILL: {blanks_list}
>
> INSTRUCTIONS:
> 1. Read the text carefully and understand the context around each blank
> 2. For each numbered blank [1], [2], [3], etc., generate a coherent paragraph that fits naturally with the surrounding text
> 3. Each generated paragraph should be substantive (at least 2-3 sentences) and maintain the style and tone of the original text
> 4. Consider the logical flow and continuity of the entire document
> 5. Output your generated paragraphs in the exact format shown below
> 6. Do not include any additional text, explanations, or comments
>
> OUTPUT FORMAT:
> 1: [Generated paragraph for blank [1]]
> 2: [Generated paragraph for blank [2]]
> 3: [Generated paragraph for blank [3]]
> (continue for all blanks)
>
> OUTPUT:"""

*Figure 22.* An example prompt for the Paragraph filling task.

You are a professional translator who can accurately preserve the meaning, tone, style, and facts of the source text when translating into English (EN). You must strictly follow English punctuation and formal written expression norms. You will keep the original formatting, paragraph structure, numbering, quotations, mathematical formulas, code snippets, and any placeholders or tags (such as [1], {VAR}, <tag>). You will not add explanations, annotations, or subjective opinions, nor will you paraphrase, expand, or omit factual information. For proper nouns such as names of people, places, or organizations, you should either use their commonly accepted English equivalents or keep the original if no standard translation exists. Do not create phonetic transliterations arbitrarily.

Please translate the following text into English (EN).
Full text: {src_text}
Target language: English (EN)
Style and rules:
 - Preserve the meaning, tone, and level of formality of the source text; follow English writing conventions and punctuation rules.
 - Keep paragraph breaks, lists, numbering, quotations, mathematical formulas, code, and any inline markers.
 - Retain and output all placeholders or tags exactly as they are (e.g., [1], {VAR}, <tag>).
 - Ensure consistency and accuracy for proper nouns, dates, numbers, and terminology; only localize units or formats when there is a clear English standard.
 - Do not add or remove factual information, invent content, or include annotations or explanations.

Instructions:
 1) Read the entire text carefully to understand the context and meaning.
 2) Translate the content into natural, fluent, and accurate English strictly following the rules above.
 3) Keep the original paragraphing and formatting; placeholders and tags must be preserved verbatim.
 4) Output only the translated text itself, without any additional content.

Output format: [Paste your English translation here]

*Figure 23.* An example prompt for the Translation task.

You are an expert summarizer who writes faithful, concise, and well-structured summaries that preserve the original text's key claims, evidence, and conclusions without adding new information.

Summarize the following document.

FULL TEXT:{full_text}
SUMMARY REQUIREMENTS:
     - Faithfulness: No new facts; keep numerical values and named entities accurate.
     - Coverage: Include the central thesis, 3–5 most important supporting points, and any conclusions/implications.
     - Clarity: Prefer unambiguous, non-redundant sentences.

INSTRUCTIONS:
 1. Identify main objective and top supporting arguments/evidence.
 2. Compress without losing essential meaning or key qualifiers.
 3. Keep the style and register requested.
 4. Do not include meta commentary.

OUTPUT FORMAT (plain text only):[Generated summary here]

*Figure 24.* An example prompt for the Summarization task.

# G. Comprehensive Analysis

## G.1. Granularity

**Word**   In the Single-QA task, models exhibit uniformly high performance. Multi-Paragraph QA also achieves consistently high accuracy, for several models it even surpasses the single-paragraph setting, indicating that LLMs can aggregate cues across paragraphs to more reliably select the correct answer.

**Sentence**   Blank omissions are a major source of error. Rates peak when the blank appears near the beginning of the document and decline monotonically toward the end shown in Figure 25. In our evaluation setting, the instruction and sentence candidate options are appended after the document, placing the decision anchor at the end. Prior work shows that long-context language models underweight distant evidence due to recency-weighted attention, and that moving salient segments closer to the decoding point mitigates this effect (Peysakhovich & Lerer, 2023). Moreover, studies on positional calibration and controlled placement indicate that performance is sensitive to where the relevant evidence sits relative to the anchor (Hsieh et al., 2024b; Xu et al., 2024). Therefore, Early blanks maximize the anchor–evidence distance that makes LLMs error. Following the ablation in Section 4.4, we relocate instructions, and find that it significantly reduces omissions at the corresponding position, which further confirm our explanation.

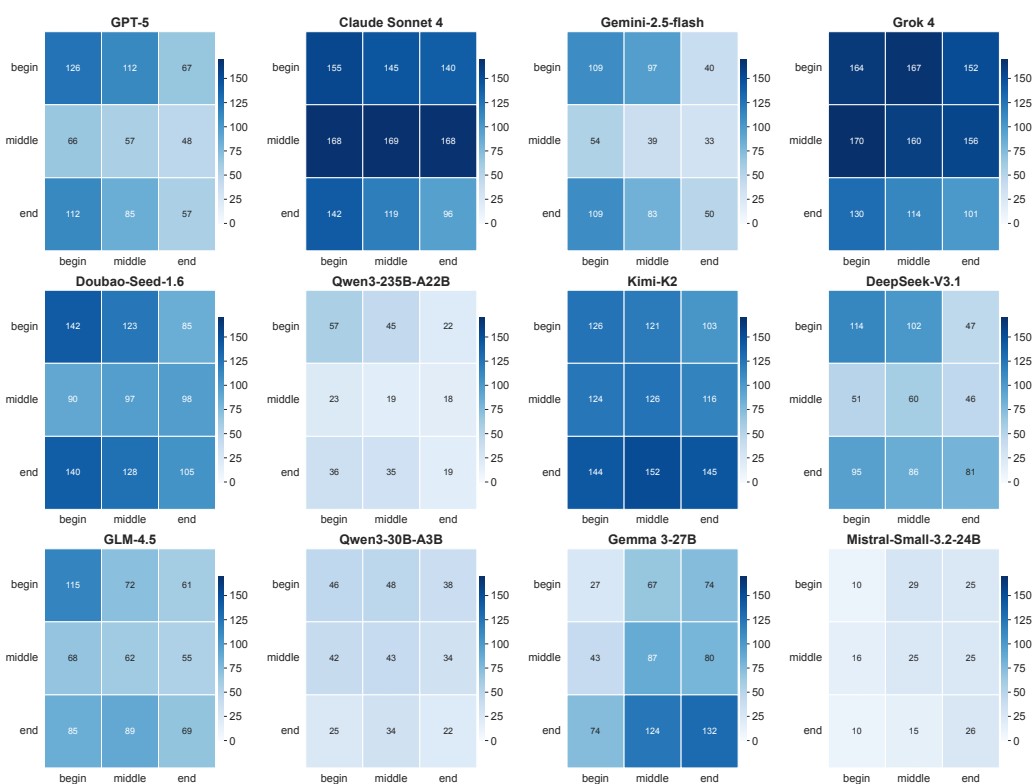

*Figure 25.* Blank omissions heatmap by position across models.

Most sentence-level errors in the generated options arise from local semantic crowding around the blank. Under local semantic crowding, where neighboring sentences share topics and entities, models preferentially follow surface cues(e.g., connectives or repeated entities) over the discourse role that the sentence plays in the surrounding argument, such as background, explanatory, or outcome.

For example, given an opening sentence like "The Working Group on the Universal Periodic Review, established in accordance with Human Rights Council resolution 5/1 of 18 June 2007, held its first session from 7 to 18 April 2008.", the correct next sentence should be "At its 15th meeting held on 16 April 2008, the Working Group adopted the present report on Algeria." (a development sentence). However, models often prefer a summary-style sentence such as "The review of Algeria was held at the 11th meeting on 14 April 2008.", which appears plausible due to repeated years and entities but serves the wrong discourse role, redundantly restating rather than advancing the argument. The position choices heatmap

with the ground truth and model predictions are in Figure 26.

We further analyze failure modes using LLM and human annotations to identify the most common error categories in model predictions in Appendix G.2.

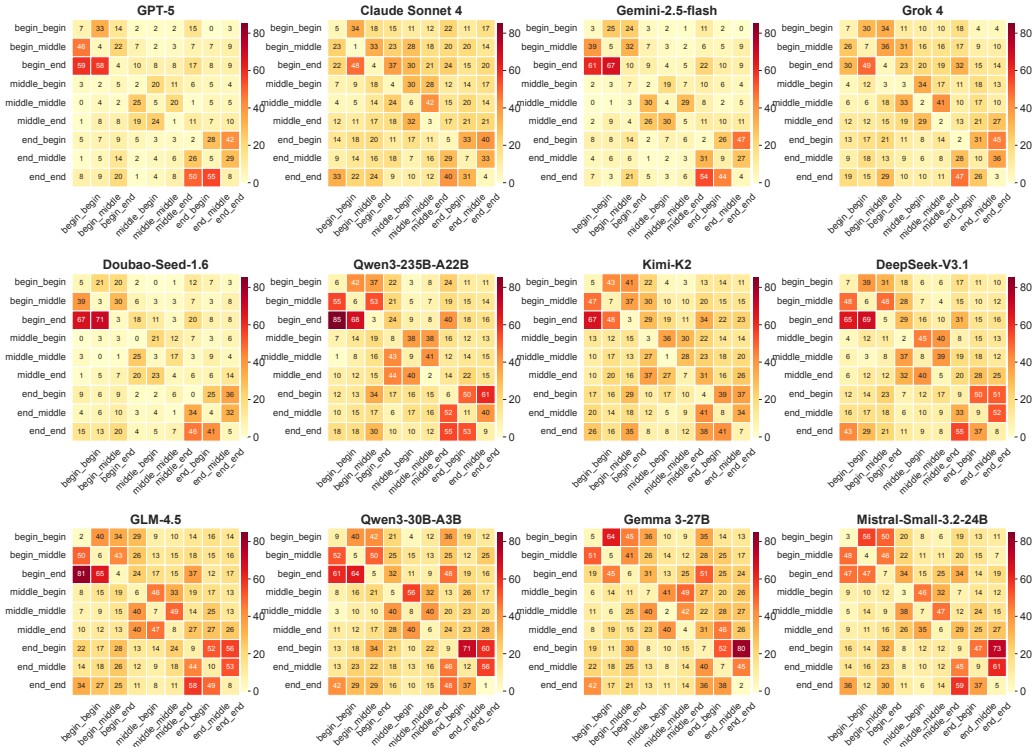

*Figure 26.* Position choices heatmap with the ground truth and model predictions.

Beyond aggregate accuracy, we observe systematic option reuse illustrated in Figure 27. Within a same cloze task, some models repeatedly select the same choice, with selections concentrated on early position: A and B more than C and D, and far more than later options. This pattern indicates an early-position bias and suggests reliance on option-frequency priors (Pezeshkpour & Hruschka, 2024b; Zheng et al., 2023a) rather than item-specific evidence under uncertainty. An ablation that shuffles option order to test in Section 4.4.

**Paragraph** Across models, generated paragraphs show a systematic bias relative to the reference: they mimic the document's surface register while underweighting diagnostic cues in adjacent paragraphs, yielding unsupported entity assertions and drift from the source's explanatory trajectory. Faithful reconstruction relies less on global paraphrase and more on local entailment with neighboring paragraphs and selective aggregation of proximal evidence(Maynez et al., 2020).

We also observe brittle local coherence: entity states and temporal anchors are not consistently propagated across adjacent context, producing continuations that read fluently yet remain locally ungrounded. For example, models may overlook concrete cues in the surrounding paragraphs and confidently assert that a policy has already been adopted when the source text only outlines actionable recommendations; they may also hallucinate on generating actors, dates, or institutions that are not present in the document. The generated text is stylistically consistent and reads smoothly, but it is factually inconsistent with the document. This illustrates the gap between fluency and consistency in long-context generation.

**Document** Models underperform on MGAL summarization: expert summaries serve as scoping prefaces that set the mandate, framing, and outline-level topic coverage while constraining claims to verifiable source content. By contrast, model outputs drift into substantive synthesis, reorder themes, enumerate cases, and import extra textual detail. The generated summaries assert trends or policy shifts without in-context support, resulting in abstractive drift and hallucinations. These patterns explain why LLM-generated summaries, though superficially fluent, align poorly with expert references, revealing a persistent gap between fluency and consistency in generated outputs.

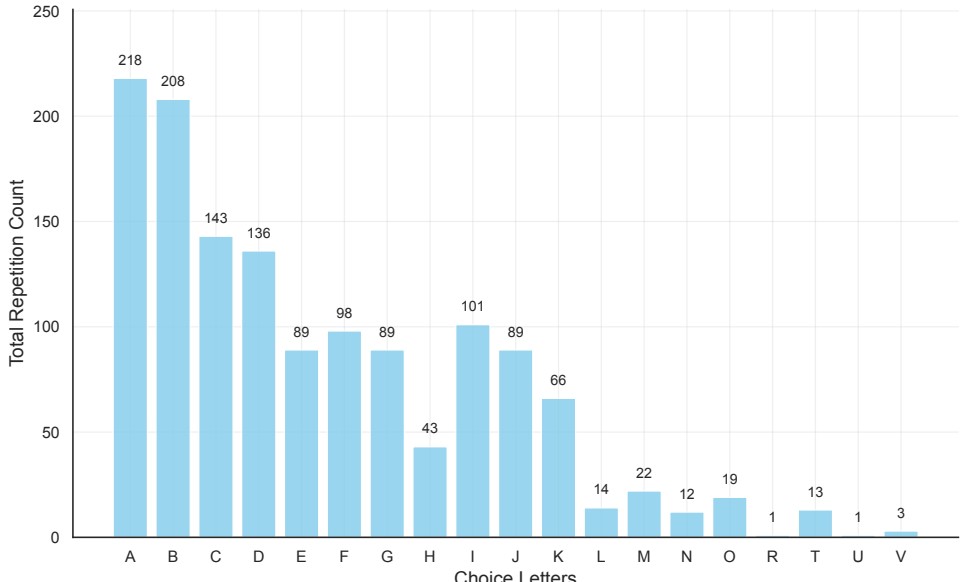

*Figure 27.* Repetition letter distribution.

In translation, most models show a higher-resource advantage, scoring consistently higher on English and other higher-resource languages than on lower-resource ones. Moreover, the GPT-5 and Gemini-2.5-flash exhibit strong performance in the Translation task among all models evaluated, significantly outperforming the average translation score of 16.85.

**Cross-granularity synthesis and implications**    From a layered analysis across four coherence-aligned linguistic granularities, we find that long-context LLMs perform well in word-level, but struggle in coarser-grained tasks. Long-context LLMs capture global semantics but struggle to recognize and exploit fine-grained discourse roles. While errors at finer units accumulate and propagate to sentences, paragraphs, and documents, yielding outputs that are superficially fluent yet weakly consistent: intra-paragraph organization is loose, and sentence-level commitments are unstable, ultimately depressing overall performance.

### G.2. Generated Options Analysis

We use GPT-5 and Gemini-2.5 to categorize sentence-level option-selection errors in Cloze with a single shared, reference-aware prompt. Each judge reads the local paragraph context and the two candidate sentences: the paragraph immediately before the target sentence; the current paragraph containing the correct sentence; and the paragraph immediately after, then returns a strict verdict. The taxonomy has three mutually exclusive classes:

- Discourse-role underuse (M1). In local semantic crowding, the sentence chosen from the model does not fill the reasoning role implied by the surrounding chain (e.g., "Given $\alpha \Rightarrow \beta$," "Although $\alpha$, still $\beta$," "$\beta$ because $\alpha$"). It functions as a parallel, rephrasing, or background statement, whereas the gold sentence uniquely completes the required role.

- Surface-cue and heuristic overuse (M2). The model-picked sentence is better explained by surface signals—explicit connectives (e.g., "however," "therefore"), repeated entities, high lexical overlap, length, or format than by context fit or entailment from the local context.

- Context or knowledge deficit (M3). The model-picked sentence conflicts with constraints evident, such as topic, entities, time, causal direction, or presupposes knowledge unsupported by the given context.

We also record secondary facets to aid diagnosis: S1 discourse-role misidentification (reserved for M1), S2 coreference drift, S3 connective or overlap lure, S4 temporal or ordinal misread, S5 negation or scope error, S6 locally coherent but globally incoherent, S7 world-knowledge gap, and S8 format or fluency bias.

Each case receives exactly one main class; secondary tags may be added. In a single pass, the judge prioritizes M1 if the gold uniquely fills the role, defaults to M2 when the choice is best explained by surface cues, and otherwise assigns M3. Ambiguity is resolved conservatively with reduced confidence.

The results are illustrated in Figure 28. We categorize the Cloze errors using the LLMs judging protocol and find a highly skewed distribution in which M1 dominates, especially in the S3 subset.

This distribution shows that the dominant failure pattern is not simply that models are attracted by surface overlap in isolation. Rather, models often choose a locally plausible sentence that shares topic words, entities, or connectives with the surrounding context, but fails to satisfy the discourse role required by the blank within the paragraph. In crowded local contexts, neighboring candidate sentences can be semantically close at the lexical or entity level, while differing in their discourse function, such as background, elaboration, contrast, evidence, or conclusion.

This suggests our finding that under local semantic crowding—adjacent sentences sharing topics and entities—models over-follow surface cues (connectives, lexical overlap, formatting) while underusing discourse-role reasoning, failing to fill the role-slot in patterns such as "Given $\alpha \Rightarrow \beta$", "Although $\alpha$, still $\beta$", or "$\beta$ because $\alpha$". This cue-following induces systematic mislabeling that mid-paragraph background or result sentences are treated as openings, and post-development summaries are misread as first sentences, because neighboring sentences share topic and register, forming deceptively plausible decoys.

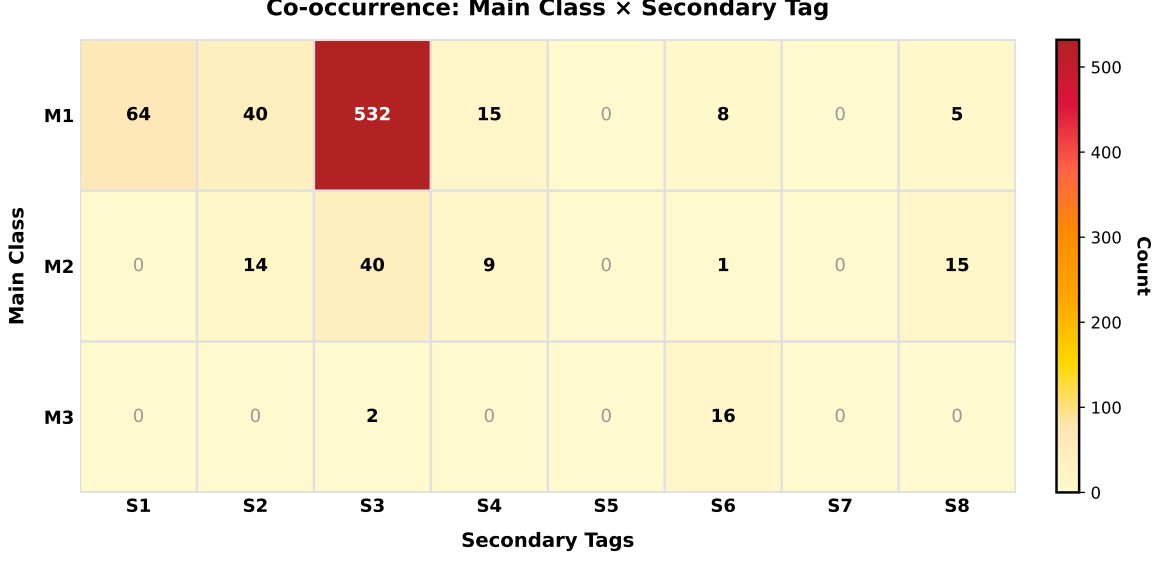

*Figure 28.* Sentence analysis results heatmap.

As shown in Table 4, we select cases in which the majority-selected wrong sentence comes from the same paragraph and is topically or lexically close to the gold sentence, but occupies a different discourse role. These cases are therefore intended to illustrate local semantic crowding rather than arbitrary distractor errors.

*Table 4.* Case studies of non-repetition local semantic crowding. The blank position is marked as __[#]__.

**Case 1: Regional human rights initiatives**

**Paragraph with blank.**    The extent of cooperation among government departments, as well as among Governments, academia, national human rights institutions and NGOs in any given country appears particularly significant. __[#]__ Examples mentioned include the Inter-American Institute of Human Rights, the European Union, the Organization for Security and Cooperation in Europe, the Council of Europe and the Secretariat of the Pacific Community.
**Gold sentence.**    Similarly, there seem to be an increasing number of regional initiatives, which States can both contribute to and benefit from, thereby fostering a valuable exchange of experiences and cross-fertilization of ideas.
**Majority-selected wrong sentence.**    Examples mentioned include the Inter-American Institute of Human Rights, the European Union, the Organization for Security and Cooperation in Europe, the Council of Europe and the Secretariat of the Pacific Community.
**Error analysis.**    The wrong sentence is locally plausible because it lists concrete regional organizations related to the same human-rights cooperation topic. However, the blank requires a general transition from national cooperation to regional initiatives before the examples are introduced. Models therefore confuse a concrete example sentence with the higher-level framing sentence required by the paragraph.

**Case 2: Gender dimension in HIV policies**

**Paragraph with blank.**    __[#]__ Member States have continued to use two approaches to address the gender dimension in HIV policies and action plans. The UNDP/World Bank/UNAIDS HIV Mainstreaming Programme supported Namibia, Botswana, Lesotho and Swaziland in strengthening their development planning processes to better integrate and implement HIV and gender priorities.
**Gold sentence.**    A joint UNAIDS/Global Fund to Fight AIDS, Tuberculosis and Malaria mission to China focused on the priority actions to be taken in the area of women and girls in China's national HIV strategy 2011–2015.
**Majority-selected wrong sentence.**    The UNDP/World Bank/UNAIDS HIV Mainstreaming Programme supported Namibia, Botswana, Lesotho and Swaziland in strengthening their development planning processes to better integrate and implement HIV and gender priorities.
**Error analysis.**    The wrong sentence is topically close because it also concerns HIV, gender priorities, UNAIDS-related support, and national policy planning. However, the paragraph first requires an opening example before summarizing the two approaches and then introducing another programme example. The error shows that models confuse a later supporting example with the opening example needed to establish the paragraph sequence.

**Case 3: Belarus electoral process**

**Paragraph with blank.**    The systemic shortcomings, such as the absence of a free system of registration for any movement, including political parties, the lack of equal access to the media by all political streams, the absence of transparency in turn out and vote counting and registration of voters and the ongoing harassment and discrimination of unwelcome candidates, render the entire electoral process not compatible with the concept of elections that are respectful of human rights and therefore pose the question of the purpose of such a process. He reiterates his readiness to work with the Government and continues to offer his support to civil society. The absence of real changes in the practices of the State apparatus and in the legal framework, notwithstanding a ready-to-implement list of recommendations, demonstrates a lack of political will to adhere to rights that are universally recognized. __[#]__ The Special Rapporteur regrets that Belarus did not take into account the numerous recommendations made by the OSCE Office for Democratic Institutions and Human Rights, the United Nations human rights mechanisms and by himself on several occasions to tackle the systemic deficiencies that underpin the electoral process in Belarus and hinder the exercise by citizens of their basic freedoms.
**Gold sentence.**    He will continue to request an official visit to the country during 2017.
**Majority-selected wrong sentence.**    He reiterates his readiness to work with the Government and continues to offer his support to civil society.
**Error analysis.**    The wrong sentence shares the same subject, cooperative stance, and Special Rapporteur-centered discourse, making it locally plausible. However, it has already functioned as a general statement of readiness. The blank instead requires a forward-looking action about requesting an official visit. Models confuse a repeated stance with the next concrete action in the sequence of recommendations.

*Table 5.* Error breakdown for the sentence-cloze task. Omission denotes failing to fill the blank with any correct sentence; Position Error denotes selecting a correct sentence from the wrong document position; Distractor denotes selecting an incorrect distractor option.

| Model | Correct | Omission | Position Error | Distractor |
|---|---|---|---|---|
| GPT-5 | 52.89 | 17.43 | 21.51 | 8.17 |
| Gemini-2.5-Flash | 52.87 | 14.66 | 22.39 | 10.08 |
| Doubao-Seed-1.6 | 47.25 | 24.06 | 21.72 | 6.96 |
| Qwen3-235B-A22B | 43.92 | 6.54 | 40.37 | 9.17 |
| Mistral-Small-3.2-24B | 40.57 | 4.32 | 40.34 | 14.77 |
| Qwen3-30B-A3B | 37.76 | 7.93 | 44.95 | 9.36 |
| DeepSeek-V3.1 | 37.75 | 16.28 | 37.86 | 8.11 |
| GLM-4.5 | 34.79 | 16.14 | 43.38 | 5.70 |
| Grok-4 | 29.17 | 31.37 | 31.96 | 7.50 |
| Kimi-K2 | 27.27 | 27.62 | 38.08 | 7.03 |
| Claude Sonnet 4 | 25.47 | 31.08 | 34.50 | 8.95 |
| Gemma-3-27B | 24.09 | 16.90 | 44.83 | 14.18 |

## G.3. Two-Level Position Analysis and Cloze Error Breakdown

**Two-level position indexing.** Figure 4 in the main paper reports document-level positional results across tasks, where each instance is grouped by whether the target evidence appears at the beginning, middle, or end of the document. For word-level QA and paragraph-level filling, position is defined only at the document level. The two-level position design is introduced specifically for the sentence-cloze task, where each removed target sentence is indexed by both its document-level position and its sentence position within the paragraph. The construction details are described in Appendix F and illustrated in Figure 6. To further disentangle these two positional dimensions for cloze, Figure 26 presents a heatmap of ground-truth versus predicted positions, and Table 7 reports cloze results at both the document and paragraph levels.

**Cloze error breakdown.** To better understand model failures in the sentence-cloze task, we further analyze the error breakdown across models. We divide predictions into four categories: *Correct*, *Omission*, *Position Error*, and *Distractor*. *Omission* denotes failing to fill the blank with any correct sentence; *Position Error* denotes selecting a correct sentence from the wrong position in the document; and *Distractor* denotes selecting an incorrect distractor option.

As shown in Table 5, distractor error rates remain relatively stable across models, generally within 5–15%. We do not observe a pattern in which non-GPT models disproportionately select GPT-4-generated distractors. Instead, the dominant error types are position errors and omissions. For example, several open-source models, including Qwen3-30B-A3B, Gemma-3-27B, and GLM-4.5, exhibit high position-error rates, while models such as Grok-4, Claude Sonnet 4, and Kimi-K2 show relatively high omission rates. This suggests that sentence-cloze failures are mainly associated with positional sensitivity and local semantic crowding, rather than systematic bias introduced by GPT-4-generated distractors.

Taken together, the two-level position analysis and the error breakdown indicate that the cloze task primarily challenges models' ability to select a sentence that is not only semantically plausible but also positionally and discourse-functionally appropriate. These results support our interpretation that model failures are driven more by positional bias and local semantic crowding than by distractor-generation artifacts.

## G.4. Context Memorization Analysis

To examine whether models truly use the provided long-context rather than relying solely on pre-training priors, we conduct a context ablation on the Single-QA task. Specifically, we compare two settings: (1) the standard setting with access to the full document context, and (2) a no-context setting where the document is withheld and the model must answer using only internal knowledge. We adopt the same evaluation setup as the Single-QA task and report results averaged over 12 LLMs.

To investigate the model's reliance on parametric knowledge versus contextual information, we conducted an context memorization study. The model average accuracy drops sharply from 0.73 (with context) to 0.31 (without context), demonstrating that performance is largely attributable to information drawn from the input rather than memorization.

We find that the model successfully answers questions related to well-known public facts, such as the "2030" timeline for malaria targets and the specific designation of a widely-known UN document. However, performance collapses when

questions require information specific to the text. For instance, the model failed to identify "2016" as the starting point for a strategic guidance initiative, defaulting instead to the more commonly known "2030 Agenda". Similarly, without the text explicitly stating "educational and financial obstacles," the model incorrectly generalized the number of obstacles as "many" instead of the correct answer "two" in the original document.

The findings show that questions grounded in commonsense or general policy remain moderately answerable without context, while those requiring document-specific details degrade substantially.

# H. Full Results on MGAL

*Table 6.* Model Performance Across All Tasks and Languages

| Task | Model | English | Chinese | Spanish | French | Russian | Arabic |
|---|---|---|---|---|---|---|---|
| **Single-QA** | GPT5 | 81.25 | 78.33 | 80.83 | **82.29** | **77.50** | **78.54** |
| | Claude Sonnet 4 | 72.75 | **78.95** | 70.00 | 74.38 | 66.88 | 71.88 |
| | Gemini-2.5-flash | 79.16 | 75.42 | 78.54 | 78.54 | 73.75 | 76.67 |
| | Grok 4 | 78.75 | 76.25 | 78.75 | 79.17 | 62.5 | 77.29 |
| | Doubao-Seed-1.6 | 78.54 | 75.83 | 78.75 | 81.46 | 68.75 | 76.46 |
| | Qwen3-235B-A22B | **82.29** | 53.75 | 77.71 | 77.71 | 66.04 | 73.54 |
| | Kimi-K2 | 78.95 | 73.33 | 78.54 | 73.13 | 67.08 | 75.42 |
| | DeepSeek-V3.1 | 79.79 | 74.79 | 76.04 | 79.79 | 66.04 | 76.88 |
| | GLM-4.5 | 76.04 | 68.96 | 75.42 | 78.75 | 61.25 | 53.33 |
| | Qwen3-30B-A3B | 73.33 | 53.54 | 75.42 | 78.75 | 67.00 | 70.63 |
| | Mistral-Small-3.2-24B | 74.79 | 60.42 | 73.33 | 73.75 | 61.25 | 71.46 |
| | Gemma-3-27B | 72.29 | 66.88 | 67.92 | 67.92 | 59.79 | 65.42 |
| **Multi-QA** | GPT5 | 82.71 | 56.88 | 62.5 | 60.42 | 54.17 | 67.29 |
| | Claude Sonnet 4 | 79.38 | 53.75 | 56.25 | 57.92 | 47.08 | 60.21 |
| | Gemini-2.5-flash | 77.08 | 56.67 | 59.58 | 62.29 | 53.75 | 59.58 |
| | Grok 4 | 78.54 | 55.00 | 37.71 | 54.79 | 39.58 | 40.21 |
| | Doubao-Seed-1.6 | 72.29 | 57.08 | 56.67 | 58.96 | 45.83 | 52.92 |
| | Qwen3-235B-A22B | 82.29 | 44.58 | 45.83 | 59.58 | 56.67 | 50.83 |
| | Kimi-K2 | **83.33** | 49.58 | 53.75 | 52.71 | 46.04 | 49.58 |
| | DeepSeek-V3.1 | 78.96 | 50.21 | 52.08 | 45.00 | 53.75 | 53.75 |
| | GLM-4.5 | 72.08 | 48.96 | 56.67 | 50.21 | **61.25** | **70.83** |
| | Qwen3-30B-A3B | 78.54 | 44.58 | 52.08 | 52.08 | 47.08 | 52.5 |
| | Mistral-Small-3.2-24B | 80.21 | **73.13** | 75.42 | 72.92 | 53.33 | 70.42 |
| | Gemma-3-27B | 78.33 | 71.25 | 67.08 | 68.13 | 59.38 | 63.13 |
| **Cloze** | GPT5 | **52.89** | 26.39 | 26.14 | 26.56 | **25.34** | 26.38 |
| | Claude Sonnet 4 | 25.47 | 19.91 | 20.48 | 20.82 | 21.61 | 20.88 |
| | Gemini-2.5-flash | 52.87 | 20.70 | 20.01 | 20.10 | 20.80 | 21.49 |
| | Grok 4 | 29.17 | 19.93 | 20.29 | 20.89 | 17.66 | 21.54 |
| | Doubao-Seed-1.6 | 47.25 | 14.59 | 14.24 | 13.78 | 13.81 | 13.13 |
| | Qwen3-235B-A22B | 43.92 | 16.49 | 16.06 | 15.87 | 13.47 | 15.98 |
| | Kimi-K2 | 27.27 | 8.33 | 9.32 | 9.32 | 7.78 | 9.95 |
| | DeepSeek-V3.1 | 37.75 | 13.83 | 14.59 | 14.03 | 9.40 | 14.28 |
| | GLM-4.5 | 34.79 | **33.18** | 12.27 | 11.25 | 9.48 | 10.83 |
| | Qwen3-30B-A3B | 37.76 | 11.36 | 12.84 | 12.69 | 11.55 | 13.04 |
| | Mistral-Small-3.2-24B | 40.57 | 21.90 | **42.90** | **42.57** | 6.78 | **39.35** |
| | Gemma-3-27B | 24.09 | 13.52 | 25.43 | 25.33 | 10.11 | 23.73 |
| **Filling** | GPT5 | 17.84 | 14.39 | 17.92 | 18.79 | 3.46 | 14.43 |
| | Claude Sonnet 4 | 20.23 | 15.23 | 3.63 | 20.60 | 1.68 | 15.49 |
| | Gemini-2.5-flash | 20.05 | **16.70** | 19.65 | 20.76 | 2.90 | 15.76 |
| | Grok 4 | **28.82** | 15.88 | **25.85** | **24.23** | 4.10 | **20.51** |
| | Doubao-Seed-1.6 | 19.67 | 16.36 | 18.75 | 19.22 | 2.76 | 14.59 |
| | Qwen3-235B-A22B | 18.43 | 15.00 | 18.63 | 18.98 | 3.60 | 13.67 |
| | Kimi-K2 | 17.26 | 13.03 | 17.09 | 18.40 | 3.13 | 13.23 |
| | DeepSeek-V3.1 | 18.23 | 15.28 | 16.72 | 19.40 | 3.77 | 13.86 |
| | GLM-4.5 | 20.95 | 16.61 | 19.49 | 20.76 | 3.64 | 15.44 |
| | Qwen3-30B-A3B | 17.54 | 13.89 | 17.33 | 17.92 | 3.33 | 12.28 |
| | Mistral-Small-3.2-24B | 18.72 | 15.98 | 19.11 | 19.95 | 3.25 | 13.76 |
| | Gemma-3-27B | 17.20 | 15.08 | 18.57 | 19.25 | **4.14** | 13.76 |
| **Summary** | GPT5 | 13.17 | 16.73 | 15.92 | 16.29 | 16.04 | 14.64 |
| | Claude Sonnet 4 | 16.56 | 27.36 | 20.82 | 21.65 | 20.43 | 25.72 |
| | Gemini-2.5-flash | 17.64 | 33.06 | 22.66 | 23.51 | 20.95 | 25.55 |
| | Grok 4 | 16.42 | 23.94 | 22.92 | 22.7 | 12.61 | 14.97 |
| | Doubao-Seed-1.6 | 16.51 | 29.5 | 18.43 | 18.74 | 11.07 | 8.21 |
| | Qwen3-235B-A22B | 19.13 | 35.53 | 22.24 | 23.15 | 21.39 | 24.17 |
| | Kimi-K2 | 16.41 | 24.02 | 19.36 | 20.53 | 14.78 | 15.93 |
| | DeepSeek-V3.1 | **22.14** | **36.61** | 24.05 | 24.47 | **23.12** | 26.53 |
| | GLM-4.5 | 21.13 | 35.34 | **26.6** | **28.71** | 21.19 | **28.3** |
| | Qwen3-30B-A3B | 17.63 | 30.32 | 23.56 | 23.44 | 17.33 | 24.16 |
| | Mistral-Small-3.2-24B | 21.8 | 2.69 | 4.76 | 3.37 | 1.55 | 2.28 |
| | Gemma-3-27B | 21.48 | 2.9 | 4.03 | 2.91 | 1.57 | 2.54 |
| **Translation** | GPT5 | 36.57 | 37.16 | 36.65 | 32.03 | 33.98 | 35.44 |
| | Claude Sonnet 4 | 21.58 | 23.07 | 9.83 | 10.97 | 12.49 | 11.86 |
| | Gemini-2.5-flash | **38.92** | 31.97 | **39.65** | **35.32** | **34.25** | **36.92** |
| | Grok 4 | 35.52 | **39.32** | 35.85 | 32.52 | 11.05 | 32.04 |
| | Doubao-Seed-1.6 | 5.88 | 4.55 | 12.82 | 5.93 | 1.64 | 6.14 |
| | Qwen3-235B-A22B | 24.25 | 13.51 | 7.23 | 4.37 | 3.04 | 4.86 |
| | Kimi-K2 | 4.47 | 5.36 | 5.47 | 5.78 | 1.62 | 4.09 |
| | DeepSeek-V3.1 | 17.46 | 36.96 | 35.23 | 30.61 | 17.40 | 33.46 |
| | GLM-4.5 | 25.34 | 23.19 | 25.32 | 22.99 | 10.41 | 21.65 |
| | Qwen3-30B-A3B | 29.98 | 14.32 | 8.21 | 6.90 | 3.94 | 2.52 |
| | Mistral-Small-3.2-24B | 1.12 | 0.59 | 2.63 | 2.91 | 0.14 | 2.38 |
| | Gemma-3-27B | 4.66 | 0.28 | 1.97 | 2.12 | 0.77 | 1.89 |

*Table 7.* Model Performance on Multi-QA Task at Document-Level.Rows indicate the position of the first paragraph and columns indicate the position of the second, with each cell representing the performance for the corresponding positional pair (e.g., 'Begin-Middle').

| Task | Model | Begin | Middle | End |
|------|-------|-------|--------|-----|
| | GPT5 | **80.42** | **78.85** | 80.21 |
| | Claude Sonnet 4 | 75.44 | 77.43 | 71.75 |
| | Gemini-2.5-flash | 77.4 | 75.94 | 77.71 |
| | Grok 4 | 77.19 | 78.75 | **80.31** |
| | Doubao-Seed-1.6 | 76.88 | 76.62 | 76.88 |
| | Qwen3-235B-A22B | 72.4 | 69.27 | 71.35 |
| Single-QA | Kimi-K2 | 76.67 | 74.17 | 72.4 |
| | DeepSeek-V3.1 | 76.67 | 74.27 | 75.73 |
| | GLM-4.5 | 72.08 | 64.38 | 66.88 |
| | Qwen3-30B-A3B | 71.75 | 67.29 | 70.18 |
| | Mistral-Small-3.2-24B | 72.17 | 69.51 | 70.26 |
| | Gemma-3-27B | 70.04 | 63.75 | 68.03 |

*Table 8.* Model Performance in Single-QA Task at Document-Level.Accuracy on the Single-QA task, categorized by the position (Begin, Middle, End) of the single answer-bearing paragraph.

| Task | Position | Model | Begin | Middle | End |
|------|----------|-------|-------|--------|-----|
| | | GPT5 | 76.54 | **78.79** | **84.18** |
| | | Claude Sonnet 4 | **77.88** | 72.86 | 79.52 |
| | | Gemini-2.5-flash | 75.12 | 77.8 | 70.69 |
| | | Grok 4 | 72.79 | 66.08 | 69.71 |
| | | Doubao-Seed-1.6 | 66.93 | 57.44 | 74.54 |
| | | Qwen3-235B-A22B | 70.21 | 72.15 | 75.02 |
| | Begin | Kimi-K2 | 76.58 | 68.04 | 84.04 |
| | | DeepSeek-V3.1 | 73.43 | 70.65 | 79.63 |
| | | GLM-4.5 | 72.67 | 70.68 | 79.34 |
| | | Qwen3-30B-A3B | 67.25 | 68.56 | 72.12 |
| | | Mistral-Small-3.2-24B | 54.73 | 72.24 | 73.02 |
| | | Gemma-3-27B | 64.08 | 76.52 | 71.21 |
| | | GPT5 | **78.79** | 79.64 | **85.23** |
| | | Claude Sonnet 4 | 72.86 | 70.35 | 82.19 |
| | | Gemini-2.5-flash | 77.80 | **80.22** | 82.93 |
| | | Grok 4 | 66.08 | 66.78 | 72.12 |
| | | Doubao-Seed-1.6 | 67.44 | 70.75 | 69.68 |
| Multi-QA | Middle | Qwen3-235B-A22B | 72.15 | 73.31 | 74.12 |
| | | Kimi-K2 | 68.04 | 79.31 | 80.41 |
| | | DeepSeek-V3.1 | 70.65 | 71.56 | 68.79 |
| | | GLM-4.5 | 70.68 | 73.59 | 75.68 |
| | | Qwen3-30B-A3B | 68.56 | 69.23 | 64.35 |
| | | Mistral-Small-3.2-24B | 72.24 | 67.28 | 66.94 |
| | | Gemma-3-27B | 76.52 | 75.61 | 71.4 |
| | | GPT5 | **84.18** | **85.23** | 74.19 |
| | | Claude Sonnet 4 | 79.52 | 82.19 | **77.47** |
| | | Gemini-2.5-flash | 70.69 | 82.93 | 71.39 |
| | | Grok 4 | 69.71 | 72.12 | 66.5 |
| | | Doubao-Seed-1.6 | 74.54 | 69.68 | 67.83 |
| | | Qwen3-235B-A22B | 75.02 | 74.12 | 65.07 |
| | End | Kimi-K2 | 84.04 | 80.41 | 75.62 |
| | | DeepSeek-V3.1 | 79.63 | 68.79 | 63.25 |
| | | GLM-4.5 | 79.34 | 75.68 | 61.25 |
| | | Qwen3-30B-A3B | 72.12 | 64.35 | 57.98 |
| | | Mistral-Small-3.2-24B | 73.02 | 66.94 | 65.43 |
| | | Gemma-3-27B | 71.21 | 71.4 | 71.15 |

*Table 9.* Model performance on the Cloze task, evaluated by hierarchical position. We measure performance at the beginning (Begin), middle (Middle), and end (End) of the document, and for each of these sections, we further evaluate at the beginning, middle, and end of the target paragraph. Scores are reported in accuracy.Best results are marked inbold, second best results are underlined.

| Task | Document-level | Model | Paragraph-level | | |
|---|---|---|---|---|---|
| | | | Begin | Middle | End |
| Cloze | Begin | GPT5 | 30.17 | 21.34 | 25.79 |
| | | Claude Sonnet 4 | 27.15 | 19.72 | 17.50 |
| | | Gemini-2.5-flash | 26.74 | 24.54 | 21.73 |
| | | Grok 4 | 28.52 | 22.83 | 20.92 |
| | | Doubao-Seed-1.6 | 30.31 | 21.32 | 14.77 |
| | | Qwen3-235B-A22B | 27.69 | 21.43 | 18.80 |
| | | Kimi-K2 | 22.01 | 14.87 | 11.93 |
| | | DeepSeek-V3.1 | 26.65 | 18.52 | 15.48 |
| | | GLM-4.5 | 24.37 | 18.11 | 13.23 |
| | | Qwen3-30B-A3B | 30.03 | 18.15 | 14.12 |
| | | Mistral-Small-3.2-24B | **38.11** | **30.85** | **26.30** |
| | | Gemma-3-27B | 31.97 | 22.19 | 17.78 |
| | Middle | GPT5 | 35.79 | 34.55 | 31.07 |
| | | Claude Sonnet 4 | 23.97 | 22.28 | 20.90 |
| | | Gemini-2.5-flash | 31.10 | 28.31 | 27.50 |
| | | Grok 4 | 19.79 | 18.43 | 17.24 |
| | | Doubao-Seed-1.6 | 22.79 | 19.36 | 17.68 |
| | | Qwen3-235B-A22B | 22.48 | 20.87 | 20.38 |
| | | Kimi-K2 | 12.55 | 11.02 | 10.09 |
| | | DeepSeek-V3.1 | 21.03 | 19.10 | 17.11 |
| | | GLM-4.5 | 17.45 | 15.51 | 12.22 |
| | | Qwen3-30B-A3B | 16.10 | 15.57 | 14.86 |
| | | Mistral-Small-3.2-24B | **40.60** | **37.96** | **33.97** |
| | | Gemma-3-27B | 27.15 | 20.01 | 17.49 |
| | End | GPT5 | **34.35** | **30.99** | **28.84** |
| | | Claude Sonnet 4 | 22.97 | 22.78 | 19.75 |
| | | Gemini-2.5-flash | 27.56 | 26.20 | 23.21 |
| | | Grok 4 | 2.31 | 22.66 | 20.79 |
| | | Doubao-Seed-1.6 | 19.18 | 15.95 | 14.18 |
| | | Qwen3-235B-A22B | 18.88 | 16.49 | 15.84 |
| | | Kimi-K2 | 8.77 | 8.17 | 7.26 |
| | | DeepSeek-V3.1 | 13.71 | 13.04 | 11.65 |
| | | GLM-4.5 | 11.30 | 10.89 | 9.80 |
| | | Qwen3-30B-A3B | 13.79 | 12.83 | 11.72 |
| | | Mistral-Small-3.2-24B | 29.21 | 27.43 | 23.33 |
| | | Gemma-3-27B | 18.65 | 13.94 | 12.51 |

## H.1. Granularity-wise Performance

### H.1.1. WORD-LEVEL SUBTASKS

*Table 11.* Performance of various models on different types of word-level question-answering tasks.On word-level question-answering tasks, GPT-5 achieves the highest scores in Numerical, Reference, and Synthesis. Gemini-2.5-flash leads in Classification, Qwen3-30B-A3B excels in Comparison, and Kimi-K2 delivers the best performance in Retrieval.Best results are marked inbold, second best results are underlined.

| Granularity-Tasks | Single-QA | | | Multi-QA | | |
|---|---|---|---|---|---|---|
| | **Numerical** | **Classification** | **Reference** | **Comparison** | **Retrieval** | **Synthesis** |
| GPT-5 | **73.44** | 80.73 | **85.31** | 62.66 | 76.52 | **87.01** |
| Claude Sonnet 4 | 62.89 | 80.64 | 81.05 | 66.77 | 73.84 | 82.71 |
| Gemini-2.5-flash | 67.29 | **81.56** | 82.19 | 63.9 | 68.72 | 84.64 |
| Grok-4 | 72.5 | 80.00 | 83.75 | 48.23 | 79.66 | 67.77 |
| Doubao-Seed-1.6 | 69.12 | 78.75 | 82.5 | 55.91 | 65.73 | 74.77 |
| Qwen3-235B-A22B | 57.38 | 76.38 | 75.25 | 65.34 | 81.79 | 82.29 |
| Kimi-K2 | 63.85 | 79.06 | 80.31 | 61.15 | **83.65** | 83.35 |
| DeepSeek V3.1 | 63.23 | 79.17 | 84.27 | 59.33 | 73.25 | 79.87 |
| GLM-4.5 | 61.33 | 78.24 | 80.35 | 60.97 | 65.76 | 77.08 |
| Qwen3-30B-A3B | 60.11 | 78.93 | 77.86 | **68.12** | 59.65 | 81.32 |
| Mistral-Small-3.2-24B | 55.23 | 77.26 | 79.45 | 64.17 | 72.29 | 72.29 |
| Gemma 3-27B | 52.93 | 75.81 | 73.09 | 56.24 | 67.59 | 77.48 |

### H.1.2. PARAGRAPH FILLING USING LLM-AS-A-JUDGE AND HUMAN EVALUATION

*Table 12.* Model performance across different metrics based on llm-as-a-judge in six UN languages. Headers are abbreviated as follows: Topic Fid. (Topic Fidelity), Local Coh. (Local Coherence), Entity Cons. (Entity Consistency), Instr. Foll. (Instruction Following), and Format Comp. (Format Compliance).Across various performance metrics, Gemini-2.5-flash achieves the highest overall score, also leading in Entity Consistency and Instruction Following. Grok-4 delivers the best Topic Fidelity, while Qwen3-235B-A22B is the strongest in Local Coherence.Best results are marked inbold, second best results are underlined.

| Model | Topic Fid. | Local Coh. | Entity Cons. | Instr. Foll. | Format Comp. | Overall |
|---|---|---|---|---|---|---|
| GPT-5 | 4.58 | 10.51 | 4.39 | 5.36 | 19.96 | 44.81 |
| Claude Sonnet 4 | 4.83 | 12.03 | 5.24 | 5.74 | 19.81 | 47.65 |
| Gemini-2.5-flash | 5.90 | 11.27 | **6.37** | **6.91** | 19.93 | **50.38** |
| Grok-4 | **5.94** | 11.83 | 5.89 | 6.70 | 19.86 | 50.22 |
| Doubao-Seed-1.6 | 5.08 | 11.29 | 5.11 | 5.90 | 19.92 | 47.29 |
| Qwen3-235B-A22B | 4.67 | **12.07** | 4.91 | 5.74 | **20.00** | 47.39 |
| Kimi-K2 | 4.14 | 10.51 | 3.94 | 5.19 | **20.00** | 43.79 |
| DeepSeek V3.1 | 4.46 | 9.64 | 4.62 | 5.71 | **20.00** | 44.43 |
| GLM-4.5 | 4.61 | 11.03 | 4.82 | 5.64 | 19.95 | 46.05 |
| Qwen3-30B-A3B | 3.79 | 10.34 | 4.59 | 5.11 | 19.98 | 43.80 |
| Mistral-Small-3.2-24B | 3.79 | 10.34 | 4.59 | 5.11 | 19.98 | 43.80 |
| Gemma 3-27B | 4.12 | 9.79 | 4.75 | 5.38 | 19.96 | 44.00 |
| Average | 4.68 | 10.90 | 4.93 | 5.70 | 19.95 | 46.08 |

*Table 13*. Model performance across different metrics based on human evaluation in Chinese and English. Headers are abbreviated as follows: Topic Fid. (Topic Fidelity), Local Coh. (Local Coherence), Entity Cons. (Entity Consistency), Instr. Foll. (Instruction Following), and Format Comp. (Format Compliance).Across various performance metrics, DeepSeek V3.1 achieves the highest overall score, also leading in Instruction Following. Qwen3-235B-A22B delivers the best Topic Fidelity, Local Coherence and Entity Consistency, while GPT-5 also lead in Entity Consistency and Qwen3-30B-A3B also delivers the best Local Coherence. Grok-4 and Mistral-Small-3.2-24B are the strongest in Local Coherence. Best results are marked inbold, second best results are underlined.

| Model | Topic Fid. | Local Coh. | Entity Cons. | Instr. Foll. | Format Comp. | Overall |
|---|---|---|---|---|---|---|
| GPT-5 | 5.32 | 12.73 | **6.78** | 6.51 | 19.88 | 51.22 |
| Claude Sonnet 4 | 5.26 | 12.42 | 6.54 | 6.63 | 19.84 | 50.69 |
| Gemini-2.5-flash | 5.93 | 11.89 | 6.32 | 6.28 | 19.85 | 50.27 |
| Grok-4 | 5.14 | 11.96 | 5.87 | 5.74 | **20.00** | 48.71 |
| Doubao-Seed-1.6 | 5.07 | 11.78 | 5.21 | 5.47 | 19.93 | 47.46 |
| Qwen3-235B-A22B | **6.15** | **12.86** | **6.78** | 5.83 | 19.83 | 51.54 |
| Kimi-K2 | 4.72 | 12.36 | 4.62 | 5.86 | 19.82 | 47.38 |
| DeepSeek V3.1 | 6.09 | 12.81 | 5.98 | **6.85** | 19.89 | **51.62** |
| GLM-4.5 | 4.53 | 10.73 | 4.26 | 5.11 | 19.79 | 44.42 |
| Qwen3-30B-A3B | 4.67 | **12.86** | 4.28 | 4.89 | 19.84 | 46.54 |
| Mistral-Small-3.2-24B | 3.86 | 10.39 | 3.88 | 5.12 | **20.00** | 43.25 |
| Gemma 3-27B | 4.07 | 10.43 | 3.94 | 5.26 | 19.87 | 43.57 |
| Average | 5.00 | 11.70 | 5.39 | 5.72 | 19.90 | 48.02 |

## H.1.3. MODEL PERFORMANCE ON TRANSLATION TASKS USING BLEU

Table 14. Translation tasks on English to Other languages. En-Ch means English translate to Chinese.In translation tasks, Gemini-2.5-flash delivers the best performance for English to Chinese, Spanish, French, and Russian, while Grok-4 achieves the highest score for English to Arabic.Best results are marked inbold, second best results are underlined.

| Model | En-Zh | En-Es | En-Fr | En-Ru | En-Ar |
|---|---|---|---|---|---|
| GPT-5 | 26.73 | 57.71 | 51.44 | 40.14 | 6.82 |
| Claude Sonnet 4 | 19.37 | 32.61 | 30.12 | 23.31 | 2.48 |
| Gemini-2.5-flash | **29.36** | **59.58** | **54.05** | **44.63** | 6.99 |
| Grok-4 | 28.17 | 53.38 | 51.76 | 36.60 | **7.68** |
| Doubao-Seed-1.6 | 12.86 | 5.80 | 5.13 | 4.20 | 1.41 |
| Qwen3-235B-A22B | 16.53 | 40.35 | 38.21 | 21.01 | 5.12 |
| Kimi-K2 | 12.49 | 4.00 | 2.98 | 2.23 | 0.65 |
| DeepSeek V3.1 | 18.33 | 17.56 | 6.82 | 37.28 | 7.34 |
| GLM-4.5 | 26.41 | 35.25 | 37.41 | 22.94 | 4.68 |
| Qwen3-30B-A3B | 21.83 | 47.30 | 42.03 | 32.70 | 6.05 |
| Mistral-Small-3.2-24B | 3.84 | 0.11 | 0.11 | 0.55 | 0.98 |
| Gemma3-27B | 9.69 | 0.08 | 0.06 | 13.33 | 0.12 |

Table 15. Translation tasks on Chinese to Other languages. Zh-En means Chinese translate to English.In Chinese to other language translations, Grok-4 achieves the highest scores for English (Zh-En) and Arabic (Zh-Ar). Gemini-2.5-flash leads in translations to Spanish (Zh-Es) and Russian (Zh-Ru), while GPT-5 delivers the best performance for French (Zh-Fr).Best results are marked inbold, second best results are underlined.

| Model | Zh-En | Zh-Es | Zh-Fr | Zh-Ru | Zh-Ar |
|---|---|---|---|---|---|
| GPT-5 | 57.30 | 42.67 | **46.91** | 31.56 | 7.35 |
| Claude Sonnet 4 | 44.65 | 24.00 | 30.12 | 11.86 | 3.34 |
| Gemini-2.5-flash | 16.95 | **50.68** | 45.11 | **39.38** | 7.73 |
| Grok-4 | **62.40** | 46.86 | 45.02 | 33.40 | **8.89** |
| Doubao-Seed-1.6 | 13.00 | 2.34 | 4.62 | 1.83 | 0.96 |
| Qwen3-235B-A22B | 14.96 | 15.93 | 31.30 | 3.52 | 1.81 |
| Kimi-K2 | 10.22 | 6.65 | 8.66 | 0.92 | 0.37 |
| DeepSeek V3.1 | 56.59 | 46.30 | 39.69 | 35.37 | 6.87 |
| GLM-4.5 | 28.69 | 33.47 | 31.45 | 19.58 | 2.74 |
| Qwen3-30B-A3B | 13.16 | 29.43 | 24.00 | 3.04 | 1.96 |
| Mistral-Small-3.2-24B | 1.00 | 0.40 | 0.40 | 0.53 | 0.61 |
| Gemma3-27B | 0.49 | 0.18 | 0.16 | 0.30 | 0.27 |

Table 16. Translation tasks on Spanish to Other languages. Es-En means Spanish translate to English.For Spanish to other language translations, GPT-5 is the top performer for Chinese (Es-Zh) and French (Es-Fr). Gemini-2.5-flash leads in translations to Russian (Es-Ru) and Arabic (Es-Ar), while Grok-4 achieves the best score for English (Es-En).Best results are marked inbold, second best results are underlined.

| Model | Es-En | Es-Zh | Es-Fr | Es-Ru | Es-Ar |
|---|---|---|---|---|---|
| GPT-5 | 25.20 | **33.23** | **57.56** | 36.32 | 7.86 |
| Claude Sonnet 4 | 16.19 | 15.79 | 11.92 | 10.05 | 0.87 |
| Gemini-2.5-flash | 35.91 | 31.13 | 52.11 | **48.58** | **8.88** |
| Grok-4 | **38.02** | 28.62 | 49.08 | 38.15 | 8.73 |
| Doubao-Seed-1.6 | 11.89 | 11.89 | 4.52 | 11.36 | 0.39 |
| Qwen3-235B-A22B | 8.81 | 5.10 | 3.37 | 2.97 | 1.58 |
| Kimi-K2 | 14.08 | 12.32 | 0.93 | 1.15 | 0.39 |
| DeepSeek V3.1 | 25.52 | 32.70 | 56.46 | 31.11 | 7.29 |
| GLM-4.5 | 17.22 | 27.86 | 46.32 | 16.01 | 7.53 |
| Qwen3-30B-A3B | 13.46 | 13.46 | 4.51 | 2.60 | 0.47 |
| Mistral-Small-3.2-24B | 1.08 | 11.27 | 0.54 | 0.64 | 1.01 |
| Gemma3-27B | 0.14 | 10.13 | 0.04 | 0.07 | 0.22 |

Table 17. Translation tasks on French to Other languages. Fr-En means French translate to English.Based on the provided table* for French translation tasks, Grok-4 achieves the highest score for English (Fr-En) and GPT-5 leads for Spanish (Fr-Es). Gemini-2.5-flash is the top performer for Chinese (Fr-Zh) and Russian (Fr-Ru), while DeepSeek V3.1 delivers the best results for Arabic (Fr-Ar).Best results are marked inbold, second best results are underlined.

| Model | Fr-En | Fr-Zh | Fr-Es | Fr-Ru | Fr-Ar |
|---|---|---|---|---|---|
| GPT-5 | 56.59 | 26.41 | **57.21** | 35.97 | 7.06 |
| Claude Sonnet 4 | 21.32 | 15.24 | 7.58 | 4.38 | 0.64 |
| Gemini-2.5-flash | 56.81 | **35.86** | 51.38 | **43.92** | 10.29 |
| Grok-4 | **59.44** | 25.3 | 50.06 | 35.40 | 9.04 |
| Doubao-Seed-1.6 | 11.84 | 6.25 | 4.62 | 40.83 | 0.48 |
| Qwen3-235B-A22B | 13.99 | 13.44 | 4.75 | 2.92 | 1.07 |
| Kimi-K2 | 11.25 | 11.68 | 2.79 | 1.22 | 0.42 |
| DeepSeek V3.1 | 56.59 | 33.17 | 43.07 | 31.61 | **11.70** |
| GLM-4.5 | 37.73 | 19.47 | 21.06 | 39.06 | 9.28 |
| Qwen3-30B-A3B | 12.14 | 20.93 | 4.41 | 2.48 | 1.10 |
| Mistral-Small-3.2-24B | 0.54 | 11.27 | 0.09 | 0.41 | 0.86 |
| Gemma3-27B | 0.20 | 9.37 | 0.07 | 0.10 | 0.11 |

Table 18. Translation tasks on Russian to Other languages. Ru-En means Russian translate to English.For the Russian translation tasks, Grok-4 achieves the highest scores for English (Ru-En), Spanish (Ru-Es), and Arabic (Ru-Ar). Gemini-2.5-flash leads in Chinese (Ru-Zh) and French (Ru-Fr).Best results are marked inbold, second best results are underlined.

| Model | Ru-En | Ru-Zh | Ru-Es | Ru-Fr | Ru-Ar |
|---|---|---|---|---|---|
| GPT-5 | 52.65 | 23.70 | 47.94 | 46.12 | 6.78 |
| Claude Sonnet 4 | 30.42 | 16.80 | 5.50 | 4.78 | 1.78 |
| Gemini-2.5-flash | 47.46 | **29.23** | 49.39 | **50.72** | 7.80 |
| Grok-4 | **56.28** | 27.78 | **49.93** | 44.48 | **9.51** |
| Doubao-Seed-1.6 | 8.81 | 14.36 | 3.12 | 3.64 | 0.79 |
| Qwen3-235B-A22B | 11.40 | 7.34 | 1.91 | 2.64 | 1.00 |
| Kimi-K2 | 10.06 | 9.09 | 3.12 | 1.13 | 0.18 |
| DeepSeek V3.1 | 55.47 | 26.64 | 41.32 | 34.73 | 9.14 |
| GLM-4.5 | 36.70 | 20.25 | 25.70 | 23.47 | 2.14 |
| Qwen3-30B-A3B | 9.43 | 6.48 | 6.25 | 2.44 | 0.74 |
| Mistral-Small-3.2-24B | 0.51 | 9.16 | 0.78 | 0.91 | 1.12 |
| Gemma3-27B | 0.16 | 9.03 | 0.08 | 0.04 | 0.12 |

Table 19. Translation tasks on Arabic to Other languages. Ar-En means Arabic translate to English.On Arabic translation tasks, GPT-5 leads in translations to English (Ar-En) and Spanish (Ar-Es), while Gemini-2.5-flash achieves the highest scores for Chinese (Ar-Zh), French (Ar-Fr), and Russian (Ar-Ru).Best results are marked inbold, second best results are underlined.

| Model | Ar-En | Ar-Zh | Ar-Es | Ar-Fr | Ar-Ru |
|---|---|---|---|---|---|
| GPT-5 | **49.41** | 19.63 | **32.97** | 39.01 | 28.89 |
| Claude Sonnet 4 | 44.04 | 5.19 | 4.71 | 3.52 | 4.97 |
| Gemini-2.5-flash | 35.42 | **22.87** | 32.57 | **46.95** | **33.43** |
| Grok-4 | 11.83 | 12.47 | 11.13 | 10.82 | 9.01 |
| Doubao-Seed-1.6 | 0.81 | 3.98 | 1.18 | 1.65 | 0.6 |
| Qwen3-235B-A22B | 7.63 | 3.80 | 0.55 | 1.85 | 1.38 |
| Kimi-K2 | 4.56 | 2.62 | 0.11 | 0.44 | 0.35 |
| DeepSeek V3.1 | 42.12 | 13.95 | 4.57 | 9.65 | 16.7 |
| GLM-4.5 | 26.41 | 15.14 | 0.77 | 6.06 | 2.81 |
| Qwen3-30B-A3B | 7.39 | 6.94 | 0.77 | 2.77 | 1.82 |
| Mistral-Small-3.2-24B | 0.28 | 0.19 | 0.10 | 0.01 | 0.10 |
| Gemma3-27B | 0.18 | 3.11 | 0.24 | 0.01 | 0.31 |

## H.1.4. MODEL PERFORMANCE ON TRANSLATION TASKS USING CHRF++

Table 20. Translation tasks on English to Other languages. En-Ch means English translate to Chinese.In translation tasks, Gemini-2.5-flash delivers the best performance for English to Chinese, Spanish, French, and Russian, while Grok-4 achieves the highest score for English to Arabic.Best results are marked inbold, second best results are underlined.

| Model | En-Zh | En-Es | En-Fr | En-Ru | En-Ar |
|---|---|---|---|---|---|
| GPT-5 | 46.83 | 82.51 | 79.25 | 72.27 | 3.49 |
| Claude Sonnet 4 | 37.47 | 59.70 | 58.33 | 50.36 | 2.19 |
| Gemini-2.5-flash | **53.13** | **83.21** | **81.58** | **74.79** | 3.43 |
| Grok-4 | 47.33 | 79.73 | 80.37 | 62.53 | **3.92** |
| Doubao-Seed-1.6 | 27.80 | 23.34 | 25.67 | 22.83 | 1.72 |
| Qwen3-235B-A22B | 45.09 | 66.23 | 67.48 | 42.99 | 2.65 |
| Kimi-K2 | 27.88 | 22.37 | 20.85 | 17.54 | 1.41 |
| DeepSeek V3.1 | 37.27 | 40.51 | 28.54 | 63.67 | 3.78 |
| GLM-4.5 | 48.94 | 61.55 | 64.93 | 46.26 | 2.90 |
| Qwen3-30B-A3B | 42.34 | 70.20 | 73.32 | 66.72 | 2.55 |
| Mistral-Small-3.2-24B | 2.54 | 7.18 | 6.92 | 5.76 | 1.43 |
| Gemma3-27B | 7.66 | 6.75 | 6.09 | 5.90 | 1.33 |

Table 21. Translation tasks on Chinese to Other languages. Zh-En means Chinese translate to English.In Chinese to other language translations, Grok-4 achieves the highest scores for English (Zh-En) and Arabic (Zh-Ar). Gemini-2.5-flash leads in translations to Spanish (Zh-Es) and Russian (Zh-Ru), while GPT-5 delivers the best performance for French (Zh-Fr).Best results are marked inbold, second best results are underlined.

| Model | Zh-En | Zh-Es | Zh-Fr | Zh-Ru | Zh-Ar |
|---|---|---|---|---|---|
| GPT-5 | 84.14 | 71.48 | **78.39** | 62.44 | 3.40 |
| Claude Sonnet 4 | 63.05 | 61.42 | 53.53 | 35.08 | 6.76 |
| Gemini-2.5-flash | 43.17 | **79.85** | 77.00 | **72.02** | 3.98 |
| Grok-4 | **85.64** | 77.14 | 77.42 | 58.48 | **9.31** |
| Doubao-Seed-1.6 | 37.47 | 22.68 | 29.95 | 18.23 | 1.24 |
| Qwen3-235B-A22B | 39.33 | 38.97 | 63.32 | 21.30 | 1.75 |
| Kimi-K2 | 32.98 | 23.24 | 27.46 | 11.39 | 1.01 |
| DeepSeek V3.1 | 82.07 | 73.93 | 68.80 | 65.48 | 4.44 |
| GLM-4.5 | 51.13 | 57.79 | 61.48 | 21.30 | 2.14 |
| Qwen3-30B-A3B | 38.64 | 61.00 | 53.57 | 20.37 | 2.61 |
| Mistral-Small-3.2-24B | 9.99 | 7.84 | 7.72 | 6.14 | 1.41 |
| Gemma3-27B | 9.43 | 6.69 | 5.92 | 6.72 | 1.40 |

Table 22. Translation tasks on Spanish to Other languages. Es-En means Spanish translate to English.For Spanish to other language translations, DeepSeek V3.1 is the top performer for French (Es-Fr). Gemini-2.5-flash leads in translations to Chineses (Es-Zh) and Russian (Es-Ru), while Grok-4 achieves the best score for English (Es-En) and Arabic (Es-AR).Best results are marked in bold, second best results are underlined.

| Model | Es-En | Es-Zh | Es-Fr | Es-Ru | Es-Ar |
|---|---|---|---|---|---|
| GPT-5 | 43.05 | 44.57 | 84.04 | 65.10 | 4.24 |
| Claude Sonnet 4 | 25.54 | 26.16 | 35.95 | 30.78 | 4.22 |
| Gemini-2.5-flash | 55.72 | **47.49** | 80.87 | 76.97 | 3.84 |
| Grok-4 | **61.93** | 43.75 | 79.35 | 64.12 | **4.40** |
| Doubao-Seed-1.6 | 24.83 | 20.88 | 39.91 | 18.93 | 0.89 |
| Qwen3-235B-A22B | 23.36 | 20.65 | 18.89 | 20.01 | 1.65 |
| Kimi-K2 | 21.66 | 19.68 | 10.08 | 13.41 | 0.99 |
| DeepSeek V3.1 | 41.33 | 47.01 | **84.32** | 58.44 | 4.22 |
| GLM-4.5 | 31.69 | 46.30 | 75.14 | 36.94 | 3.75 |
| Qwen3-30B-A3B | 22.07 | 21.77 | 25.03 | 19.57 | 1.60 |
| Mistral-Small-3.2-24B | 8.99 | 7.10 | 6.62 | 5.54 | 1.37 |
| Gemma3-27B | 7.80 | 7.35 | 5.65 | 5.41 | 1.30 |

Table 23. Translation tasks on French to Other languages. Fr-En means French translate to English.Based on the provided table for French translation tasks, Grok-4 achieves the highest score for English (Fr-En). Gemini-2.5-flash is the top performer for Chinese (Fr-Zh), Spanish (Fr-Es) and Russian (Fr-Ru), while DeepSeek V3.1 delivers the best results for Arabic (Fr-Ar).Best results are marked in bold, second best results are underlined.

| Model | Fr-En | Fr-Zh | Fr-Es | Fr-Ru | Fr-Ar |
|---|---|---|---|---|---|
| GPT-5 | 80.66 | 41.82 | 72.77 | 70.50 | 2.86 |
| Claude Sonnet 4 | 46.83 | 21.46 | 30.33 | 23.68 | 2.16 |
| Gemini-2.5-flash | 82.98 | **49.55** | **79.35** | **74.02** | 6.93 |
| Grok-4 | **84.32** | 38.82 | 79.01 | 64.69 | 5.60 |
| Doubao-Seed-1.6 | 36.17 | 20.48 | 25.97 | 72.94 | 1.22 |
| Qwen3-235B-A22B | 38.96 | 22.65 | 26.40 | 21.67 | 1.46 |
| Kimi-K2 | 33.58 | 22.59 | 17.07 | 13.56 | 1.03 |
| DeepSeek V3.1 | 82.96 | 45.42 | 70.93 | 58.54 | **7.98** |
| GLM-4.5 | 61.86 | 32.57 | 45.30 | 70.71 | 6.90 |
| Qwen3-30B-A3B | 37.16 | 23.72 | 25.75 | 19.83 | 1.61 |
| Mistral-Small-3.2-24B | 8.98 | 7.29 | 7.15 | 5.94 | 1.41 |
| Gemma3-27B | 8.05 | 7.29 | 6.76 | 5.83 | 1.28 |

Table 24. Translation tasks on Russian to Other languages. Ru-En means Russian translate to English.For the Russian translation tasks, Grok-4 achieves the highest scores for English (Ru-En). DeepSeek V3.1 leads in Arabic (Ru-Ar), while Gemini-2.5-flash performs best in Chinese (Ru-Zh), Spanish (Ru-Es) and French (Ru-Fr).Best results are marked in bold, second best results are underlined.

| Model | Ru-En | Ru-Zh | Ru-Es | Ru-Fr | Ru-Ar |
|---|---|---|---|---|---|
| GPT-5 | 79.72 | 39.73 | 77.12 | 73.54 | 4.01 |
| Claude Sonnet 4 | 56.28 | 22.24 | 24.48 | 26.43 | 3.95 |
| Gemini-2.5-flash | 73.30 | **46.33** | **80.83** | **81.01** | 4.18 |
| Grok-4 | **83.95** | 43.75 | 80.14 | 77.48 | 5.45 |
| Doubao-Seed-1.6 | 34.61 | 32.67 | 21.81 | 22.47 | 0.91 |
| Qwen3-235B-A22B | 33.57 | 21.27 | 19.29 | 19.79 | 0.89 |
| Kimi-K2 | 32.06 | 22.19 | 17.11 | 13.33 | 1.00 |
| DeepSeek V3.1 | 83.54 | 42.38 | 68.01 | 64.66 | **6.84** |
| GLM-4.5 | 63.25 | 32.38 | 50.36 | 50.84 | 1.83 |
| Qwen3-30B-A3B | 20.82 | 9.77 | 13.92 | 10.25 | 1.69 |
| Mistral-Small-3.2-24B | 8.68 | 6.64 | 7.01 | 6.84 | 1.36 |
| Gemma3-27B | 7.58 | 7.10 | 6.54 | 5.37 | 1.29 |

Table 25. Translation tasks on Arabic to Other languages. Ar-En means Arabic translate to English.On Arabic translation tasks, GPT-5 leads in translations to English (Ar-En), while Gemini-2.5-flash achieves the highest scores for Chinese (Ar-Zh), Spanish (Ar-Es), French (Ar-Fr), and Russian (Ar-Ru).Best results are marked in bold, second best results are underlined.

| Model | Ar-En | Ar-Zh | Ar-Es | Ar-Fr | Ar-Ru |
|---|---|---|---|---|---|
| GPT-5 | **78.37** | 40.30 | 62.50 | 72.35 | 59.28 |
| Claude Sonnet 4 | 73.20 | 19.40 | 18.32 | 21.16 | 20.78 |
| Gemini-2.5-flash | 59.20 | **43.31** | **63.49** | **83.23** | **64.35** |
| Doubao-Seed-1.6 | 12.43 | 11.31 | 13.56 | 14.94 | 8.78 |
| Grok-4 | 34.14 | 20.14 | 33.17 | 33.51 | 38.39 |
| Qwen3-235B-A22B | 31.05 | 18.70 | 1.28 | 14.12 | 16.67 |
| Kimi-K2 | 23.82 | 12.22 | 0.87 | 11.47 | 9.33 |
| DeepSeek V3.1 | 67.61 | 33.41 | 5.03 | 5.84 | 40.26 |
| GLM-4.5 | 57.19 | 33.95 | 1.52 | 42.13 | 10.99 |
| Qwen3-30B-A3B | 32.65 | 16.05 | 1.53 | 21.19 | 17.06 |
| Mistral-Small-3.2-24B | 6.37 | 1.68 | 2.74 | 3.88 | 2.44 |
| Gemma3-27B | 3.85 | 2.98 | 0.48 | 0.31 | 2.03 |

