# OpenReview forum: "MGAL: A Multilingual Granularity-Aware Long-Context Benchmark"
_ICML.cc/2026/Conference — ICML 2026 regular_

### Official Review · Reviewer_Bt7Y · 2026-02-25

**Soundness:** 3
**Presentation:** 3
**Significance:** 2
**Originality:** 3
**Overall Recommendation:** 4
**Confidence:** 4

**Summary:**

The core of this paper is not to propose a new long-context model, but rather to fill a gap in existing long-context evaluation. Many prior benchmarks focus mainly on document-level tasks and high-resource languages, making it difficult to tell at what level models truly understand long texts. To address this issue, the authors introduce MGAL, a multilingual benchmark for long-context evaluation. It is built on long documents in the six official UN languages, and it breaks evaluation down into finer levels of granularity, including words, sentences, paragraphs, and full documents. It also controls whether the key information appears at the beginning, middle, or end of the context, allowing for a more systematic analysis of how models handle positional variation and different linguistic phenomena. The benchmark covers a range of task formats, including question answering, sentence completion, paragraph completion, summarization, and translation. The dataset construction further emphasizes cross-lingual alignment and incorporates human verification to ensure quality.

**Compliance With Llm Reviewing Policy:**

Affirmed.

**Final Justification:**

To align with the authors' statement, I will retain my score.

**Key Questions For Authors:**

1. Did the evaluation metrics you selected in your experiments consider the efficiency and scalability of the method in practical applications?
2. In your experiments, did you consider the diversity of different datasets and the impact of data preprocessing? If so, could you share how these factors influenced the analysis of the results?

**Limitations:**

yes

**Strengths And Weaknesses:**

Soundness.
The paper is generally sound as a benchmark and evaluation study. The experimental setup is broad and systematic, covering multiple task granularities, languages, and 12 representative long-context models under a unified setting. The use of automatic metrics, LLM-based judgment, human verification, and an additional ablation study strengthens the empirical support. A key limitation is that metrics such as BLEU and ROUGE-L are still crude proxies for long-context generation and translation quality, so some conclusions about higher-level semantic consistency are not fully convincing. In addition, the ablation analysis is mostly concentrated on simpler tasks, leaving the role of pre-trained knowledge in paragraph- and document-level performance less clearly examined.
Presentation.
The paper is clearly written and well structured overall. The motivation, benchmark design, and evaluation pipeline are easy to follow, and the work is positioned well relative to prior long-context and multilingual benchmarks. That said, the paper is dense in places, and some analytical claims would benefit from being stated more cautiously and separated more clearly from direct empirical findings.
Significance.
The paper addresses an important problem, since current long-context evaluation remains limited in both language coverage and diagnostic depth. A benchmark that studies long-context ability across multiple linguistic levels and languages is valuable and could be useful for future model evaluation and development. The significance is somewhat bounded, however, because the contribution is primarily evaluative rather than a direct modeling advance.
Originality.
The originality lies mainly in the benchmark design rather than in a new method. The combination of multilingual coverage, granularity-aware evaluation, and positional analysis gives the work a meaningful diagnostic perspective and helps reveal behaviors that standard benchmarks may miss. At the same time, most of the individual ingredients are not new by themselves, so the novelty comes more from careful integration than from a fundamentally new idea.

---

> ### Author Rebuttal · Authors · 2026-03-31
>
> Thank you for your valuable comments.
>
> ### **Q1: The efficiency and scalability of evaluation metrics**
>
> We consider it. Our main benchmark metrics: Acc, ROUGE-L, and BLEU were chosen for practical efficiency, reproducibility, and scalability across many models, tasks, and languages. For summarization, we follow established long-context benchmarks such as ∞BENCH, CLongEval, and LongBench, which also use ROUGE-L; for translation, we use sacreBLEU as a standard reference-based metric. For the newly introduced open-ended paragraph-filling task, we additionally report LLM-as-a-judge and human evaluation to capture quality dimensions not fully reflected by ROUGE-L. Although judge-based evaluation is more costly than automatic scoring, it is still far more scalable than full human evaluation, and our carefully designed prompt makes it consistent and reproducible. The high Spearman correlation between LLM-as-a-judge and human evaluation further supports it as a practical compromise between evaluation quality and scalability. All evaluation prompts are provided in Appendix G.7.
>
> ### **Q2: Diversity of data processing and analysis**
>
> We consider it. We explicitly considered dataset diversity in the benchmark design. MGAL is built from about 70,000 publicly available UN reports, from which we identify roughly 4,000 suitable source reports for benchmark construction. Across tasks, it covers all six official UN languages with the same number of query–response pairs per language, and documents span a broad length range of 8K–128K tokens. We also consider topic diversity: the source reports span multiple UN policy and governance domains, including international development, human rights, peace and security, humanitarian affairs, public health, environment, education, and economic/social affairs. Because the corpus is large, we can sample across multiple topics while preserving similar high-level report structure, making cross-task and cross-language comparisons more controlled. As a result, the observed performance patterns are less likely to be driven by a narrow slice of data.
>
> ### **W1: Limitations of BLEU/ROUGE-L**
>
> In MGAL, we use ROUGE-L and BLEU mainly for consistency with prior long-context benchmarks and to provide a standardized comparison across models. For summarization, we follow established practice in benchmarks such as ∞BENCH, CLongEval, and LongBench, which also use ROUGE-L; for translation, we report BLEU computed with sacreBLEU as a standard reference-based metric. At the same time, we do not rely on reference metrics alone for MGAL’s newly introduced open-ended generation setting. For paragraph filling, which has no direct counterpart in prior long-context benchmarks, we additionally report LLM-as-a-judge and human evaluation to capture quality aspects not fully reflected by ROUGE-L; Appendix I.1.2 and Appendix G.5 show high Spearman correlation between them, suggesting that judge-based evaluation aligns well with human preferences on this task.
>
> ### **W2: Limited ablation beyond simpler tasks**
>
> For QA, we additionally conduct a context ablation on Single-QA (Appendix H.3) to test whether models rely on the provided document rather than parametric memory. For paragraph filling, we complement ROUGE-L with LLM-as-a-judge and human evaluation, which reveal a consistent fluency–consistency gap, especially in Topic Fidelity and Entity Consistency.
>
> ### **W3: Analytical claims should be more cautious**
>
> We will soften several analytical claims and present them as benchmark-grounded observations rather than fully established general conclusions. We will also improve the organization of the analysis section so that evidence and interpretation are more clearly distinguished.
>
> ### **W4: Primarily evaluative contribution**
>
> MGAL is primarily an evaluation benchmark rather than a modeling contribution. Our goal is to provide a controlled multilingual testbed that reveals previously underexplored long-context weaknesses across granularity and position.
>
> ### **W5: Novelty mainly comes from integration**
>
> The novelty of MGAL lies in its unified integration of multilingual alignment, granularity-aware task design, and position-aware evaluation within a single controlled long-context benchmark. We will revise the paper to state this contribution more precisely.

---

> > ### Author Rebuttal · Reviewer_Bt7Y · 2026-04-02
> >
> > To align with the authors' statement, I will retain my score.

---

> > > ### Author Response · Authors · 2026-04-04
> > >
> > > Thank you for your follow-up and for taking the time to continue engaging with our rebuttal. We appreciate your careful consideration and your decision to retain the score at this stage.
> > >
> > > Best,
> > >
> > > Authors of submission 14952

---

### Official Review · Reviewer_GobG · 2026-03-04

**Soundness:** 2
**Presentation:** 2
**Significance:** 3
**Originality:** 3
**Overall Recommendation:** 4
**Confidence:** 4

**Summary:**

The paper introduces MGAL, a multilingual benchmark designed to evaluate long-context large language models across multiple levels of linguistic granularity. The authors investigate the concept of granularity-aware evaluation and attempt to address a pertinent problem in existing long-context benchmarks, which largely focus on document-level tasks and high-resource languages.

MGAL is constructed from United Nations reports with contexts ranging from about 8K to 128K tokens across the six official UN languages. The benchmark evaluates model performance at four linguistic levels (word, sentence, paragraph, and document) and also controls for the position of evidence within the document. Tasks include question answering, sentence cloze, paragraph completion, summarization, and translation, with instances generated through an LLM-assisted pipeline followed by human verification.

Experiments on 12 long-context LLMs show that models perform relatively well on fine-grained tasks but struggle more at coarser granularities. The authors also analyze positional effects, option-order biases, and cases where models rely on shallow lexical cues rather than discourse-level reasoning.

**Compliance With Llm Reviewing Policy:**

Affirmed.

**Final Justification:**

I maintain Weak Accept after the rebuttal.

The rebuttal successfully addressed my main concerns: (1) quantitative evidence for local semantic crowding is now clearer, (2) detailed acceptance rates demonstrate rigorous human verification, and (3) the rationale for focusing ablations on sentence-cloze is well-justified.

I've increased my score accordingly but maintain "Weak Accept" because the clarity issues remain unresolved in the current manuscript. The authors promise improvements (earlier examples, clearer figures, better metric explanations) for the camera-ready version, but naturally, these aren't yet visible.

The paper makes a solid, timely contribution to long-context LLM evaluation with good empirical coverage. With the promised revisions, it will be a stronger addition to the literature.

**Key Questions For Authors:**

The paper introduces “local semantic crowding” as an explanation for model errors in the sentence-cloze task. Could the authors clarify the analysis described in Appendix H.2 and explain more explicitly what quantitative evidence supports this claim?

The dataset generation pipeline relies heavily on LLMs with human verification. Could the authors provide more detail about the types of errors or artifacts encountered during manual checks, and how frequently generated items had to be corrected or discarded?

Finally, the ablation studies focus only on the sentence-cloze task. Is there a reason why similar analyses were not conducted for the other tasks (e.g., paragraph filling or QA), and do the authors expect the observed effects to generalize across them?

**Limitations:**

yes

**Strengths And Weaknesses:**

# Strengths

1. Timely benchmark direction. The paper addresses an increasingly relevant problem: evaluating long-context LLMs beyond document-level tasks and across multiple granularities. The focus on fine-grained linguistic units within long documents fills a relatively underexplored niche in long-context evaluation.
1. Extensive empirical evaluation. The benchmark is evaluated across a broad set of models, including both open-source and proprietary systems, and across multiple languages and task types. This provides a comprehensive overview of current long-context model capabilities.
1. Interesting data construction. The use of United Nations reports provides naturally aligned multilingual documents with consistent paragraph and sentence structure across languages. This enables cross-lingually comparable tasks while controlling for discourse structure and document position.

# Weaknesses
1. Heavy reliance on LLM-generated data. The benchmark appears largely generated through LLM pipelines with humans primarily verifying outputs. While manual checks are mentioned, it is unclear whether they sufficiently address potential biases or artifacts introduced during generation. More discussion or analysis of such biases would strengthen confidence in the dataset.
1. Presentation and clarity issues. Some aspects of the benchmark are harder to understand than necessary. Concrete examples appear relatively late in the paper, which makes the task setup difficult to grasp early on. Figure 1 is also insufficiently explained (e.g., the difference between subfigures b) and c) and the roles of humans vs. LLMs in the pipeline). In addition, Table 2 mixes heterogeneous metrics (accuracy, ROUGE-L, BLEU) without clearly explaining their interpretation or scale, which gives a skewed impression of model eprformance
1. Limited support for the “local semantic crowding” claim. The discussion of this phenomenon appears largely interpretative. While the paper provides qualitative examples, it is unclear whether the claim is backed by systematic quantitative evidence. Appendix H.2 is also difficult to follow, which makes it challenging to assess how strongly the evidence supports this conclusion (s. questions - I might have misunderstood something here)
1. Analysis section is mostly qualitative. Section 5 contains several interesting observations, but most are described qualitatively. This is not an inherent weakness, but if the discussion stays qualitative, I' at least expect more illustrative examples (better: quantitative evidence) for each paragraph, to strengthen the analysis.
1. Limited language diversity and ablation coverage. Although the benchmark is multilingual, it includes the six official UN languages, which are relatively well resourced; so the claim for a coverage of "low resource languages" is debatable. Additionally, the ablation studies are restricted to the sentence-cloze task, which lacks a clear justification.

---

> ### Author Rebuttal · Authors · 2026-03-31
>
> Thank you for your valuable comments.
>
> ### **Q1&W3: explain for “local semantic crowding”**
>
> We provide quantitative validation of local semantic crowding in Appendix H.2 through a systematic error analysis of sentence-cloze mistakes using two independent LLM judges (GPT-5 and Gemini-2.5) with a shared reference-aware prompt. Each error is assigned exactly one main class M1 discourse-role underuse, M2 surface-cue/heuristic overuse, or M3 context/knowledge deficit, and may also receive secondary tags such as connective or lexical-overlap lure (S3). The key evidence is the distribution in Figure 28: errors are heavily concentrated in M1, especially M1 × S3 = 532, far exceeding M2 × S3 = 40 and M3 × S3 = 2. This shows that the dominant failure pattern is choosing a locally plausible sentence that shares topic, entities, or connectives while failing to satisfy the discourse role required by the paragraph context. This is the main quantitative basis for the local semantic crowding claim, and we will revise Appendix H.2 and the main text to make this more explicit.
>
> ### **Q2&W1: Annotation details**
>
> Human review is integrated throughout the pipeline, with rare annotator disagreement, and all question–answer pairs are checked for correctness and cross-lingual consistency. We report accepted vs. generated items (including rejections due to disagreement or quality issues): Single-paragraph 420/582, Multi-paragraph 420/604, Cloze 420/800, Paragraph-filling 420/500, Summary 420/800. In QA tasks, rejections mainly arise from semantic ambiguity or shortcut cues enabling answers without true contextual grounding, or from conflation of distinct meanings across paragraphs. In cloze and paragraph-filling, rejections are due to unbalanced selection (e.g., clustered sentences/paragraphs). In summarization, rejections stem from formatting inconsistencies in UN reports causing extraction or cross-lingual misalignment errors.
>
> ### **Q3: Ablation coverage**
>
> The ablations in Section 4.4 focus on sentence cloze because positional and decision-related effects are most directly testable there under controlled conditions. Sentence cloze uses fixed candidate options and a clearly defined decision anchor (instruction block plus answer choices), making it possible to isolate the effects of instruction placement and option order. These factors are much less cleanly disentangled in open-ended tasks such as paragraph filling or summarization, where there is no fixed answer set and generation introduces additional variability. For QA, we additionally conduct a context ablation on Single-QA (Appendix H.3) to test reliance on the provided document rather than parametric memory. For paragraph filling, we complement ROUGE-L with LLM-as-a-judge and human evaluation, which reveal a consistent fluency–consistency gap, especially in Topic Fidelity and Entity Consistency. Only the cloze task is constructed at the sentence level, so local semantic crowding arises specifically in sentence-level tasks.
>
> ### **W2: Presentation and clarity issues**
>
> We will move representative task examples earlier in the paper. In the revision, we will revise Figure 1 and its caption to more clearly distinguish subfigures, explain the roles of humans versus LLMs in the pipeline, and make metric interpretation more explicit. Figure 2 already presents the generation and human-verification pipeline, and we will connect it more clearly to the main text. The metrics in Table 2 are also discussed in Appendix G.1, where we explain that accuracy, ROUGE-L, and BLEU are not directly comparable in absolute value and should be interpreted mainly through relative trends within each task; we will clarify this in the main text.
>
> ### **W4: Analysis section is mostly qualitative**
>
> For local semantic crowding, we provide quantitative analysis in Appendix H.2. For the fluency–consistency gap, Appendix G.4, G.5, and Table I.1.2 provide supporting evidence. Due to space limits, we will add more illustrative examples in the revision.
>
> ### **W5: Limited language diversity**
>
> All six languages in our dataset are high-resource by conventional measures, with very large speaker populations: English (1.45B), Chinese/Mandarin (1.13B), Spanish (559M), French (310M), Arabic/Modern Standard (274M), and Russian (255M). Our analysis concerns relative resource imbalance within this high-resource subset, highlighting cross-language performance differences without confling them with genuinely low-resource languages; we will clarify this in the revision.

---

> > ### Author Rebuttal · Reviewer_GobG · 2026-04-01
> >
> > I feel like most of my points are addressed and I've accordingly increased my score. I still maintain with a "weak accept" as the clarity issues persist - the authors promise improvements here, but naturally this is not reflected in the current paper state yet.

---

> > > ### Author Response · Authors · 2026-04-01
> > >
> > > Thank you very much for your careful reading, constructive suggestions, and for increasing your score. We are glad that most of your concerns have been addressed.
> > >
> > > We also appreciate your point regarding the remaining clarity issues. Due to the ICML rebuttal format, we are not able to revise the main paper text directly at this stage, so these promised improvements are naturally not yet visible in the current manuscript. However, we fully agree with your suggestions and are committed to incorporating them into the final version, including clearer presentation, earlier task examples, and improved explanations of figures and metric interpretation.
> > >
> > > Best,
> > > Authors of submission 14952

---

### Official Review · Reviewer_mpQg · 2026-03-11

**Soundness:** 3
**Presentation:** 3
**Significance:** 3
**Originality:** 2
**Overall Recommendation:** 4
**Confidence:** 3

**Summary:**

The paper presents MGAL, a multilingual long-context benchmark built from UN reports in six languages. It evaluates models across four granularity levels: word, sentence, paragraph, and document, and also controls for position in the context. The main contribution is the benchmark design, not a new model. The experiments support the broad finding that current LLMs do much better on fine-grained retrieval-style tasks than on coarser generation tasks, and that closed-source models still perform better in lower-resource languages.

**Compliance With Llm Reviewing Policy:**

Affirmed.

**Final Justification:**

The rebuttal addresses several points and improves clarity, but it does not materially change my main concerns. In particular, it adds helpful detail on filtering and analysis, yet still does not provide stronger evidence of annotation reliability, and the LLM-as-a-judge validation remains somewhat limited relative to the paper's multilingual open-ended claims. The revisions promised on claim strength and domain narrowness are appropriate. Overall, my concerns are partially resolved, but not enough to justify changing my score.

**Key Questions For Authors:**

1. Can you provide stronger annotation quality evidence, such as inter-annotator agreement or adjudication statistics, instead of only saying items were approved by annotators?
2. How many distinct source reports are used, and how balanced are length, topic, and language distributions across tasks?
3. For paragraph filling and summarization, can you validate the evaluation more broadly across all six languages, not only with limited human checks?
4. How much of the observed difficulty is really about long-context reasoning, versus the fact that UN reports are very structured and stylistically regular?
5. The claims about local semantic crowding and the fluency-consistency gap are interesting, but can you support them with a more controlled analysis?

**Limitations:**

No. The paper discusses some metric and judge limitations, which is good, but it should more clearly discuss the narrow source domain. Since everything comes from UN reports, the benchmark may overrepresent formal, parallel, and highly structured discourse. It should also acknowledge more directly that some tasks may partly measure translation or summarization ability, not only long-context understanding.

**Strengths And Weaknesses:**

On soundness, this is a solid benchmark paper overall. The multilingual, granularity-aware, and position-aware design fills a real gap, and the experiments go beyond a simple leaderboard by adding analysis of failure patterns. The main weaknesses are in data quality and evaluation. Much of the benchmark is LLM-generated before human filtering, but there is little evidence about annotation reliability beyond basic approval. For open-ended tasks, the evaluation still depends a lot on imperfect automatic metrics, and the validation of LLM-as-a-judge is fairly limited. so the paper is generally sound, but some claims rely on evidence that feels a bit thinner than ideal.

on presentation, the paper is clear and easy to follow. The motivation, benchmark structure, and experiments are organized well. Still, a few claims are stated too strongly, especially the parts framed as "new challenges", which read more like plausible interpretations than fully established findings. There are also some small places where the exact task setup or counting could be clearer.

On significance, I think the paper makes a meaningful contribution. A benchmark covering six UN languages, explicit granularity levels, and position sensitivity could be useful for future work on long-context evaluation beyond English-only settings. The main limitation is domain narrowness: UN reports are formal, parallel, and highly structured, so the benchmark may not fully reflect harder real-world long-context settings.

On originality, the novelty is moderate. There is no new method, but the benchmark framing is useful and well motivated. The combination of multilingual, granularity-aware, and position-aware evaluation is a nice resource contribution, even if the individual ingredients are not entirely new.

---

> ### Author Rebuttal · Authors · 2026-03-31
>
> Thank you for your valuable comments.
>
> ### **Q1: Annotation quality evidence**
>
> Human review is integrated throughout the pipeline, with rare annotator disagreement, and all question–answer pairs are checked for correctness and cross-lingual consistency. We report accepted vs. generated items (including rejections due to disagreement or quality issues): Single-paragraph 420/582, Multi-paragraph 420/604, Cloze 420/800, Paragraph-filling 420/500, Summary 420/800. In QA, rejections mainly arise from semantic ambiguity or shortcut cues enabling answers without true contextual grounding, or from conflation of distinct meanings across paragraphs. In cloze and paragraph-filling, rejections are due to unbalanced selection (e.g., clustered sentences/paragraphs). In summarization, rejections stem from formatting inconsistencies in UN reports causing extraction cross-lingual misalignment errors.
>
> ### **Q2: distribution of the used reports**
>
> MGAL is built from a large pool of about 70,000 UN reports, from which we first identify roughly 4,000 suitable source reports for construction. Each task covers all six UN languages and contains the same number of query–response pairs per language. For document length, we sample the same number of reports across 8K to 128K tokens. In topic coverage, the source documents span diverse UN policy and governance domains, including international development, human rights, peace and security, humanitarian affairs, public health, environment, education, and economics. The UN corpus is large enough to support sampling across multiple topical areas while maintaining similar structures. We will clarify it.
>
> ### **Q3: Evaluation of paragraph-filling and summarization**
>
> Under the same LLM-as-a-judge guidelines, we supplement six-language evaluations for paragraph filling and summarization and report detailed results. The original paper in Appendix I.1.2 also includes human evaluation, and Appendix G.5 shows a high Spearman correlation between LLM and human judgments, supporting the reliability of LLM evaluation.
> The tables referred in this response can be found at this anonymous link [https://anonymous.4open.science/r/test-9057/](https://anonymous.4open.science/r/test-9057/)
>
> ### **Q4: the observed difficulty is really about long-context reasoning**
>
> To separate contextual understanding from memorization or stylistic priors, we include a context ablation in Appendix H.3 on Single-QA: when document context is removed, average accuracy across 12 LLMs drops sharply from 0.73 to 0.31, indicating that performance depends substantially on information extracted from provided long context rather than only on parametric knowledge or stylistic familiarity.
>
> ### **Q5: controlled analysis of local semantic crowding**
>
> We provide controlled analyses in Appendix H.2. For the sentence-cloze task, we go beyond aggregate accuracy and examine error patterns under controlled positional settings, including omission heatmaps and judge-based error categorization (Figures 26 and 28). These analyses show that errors concentrate where adjacent sentences share topics, entities, and discourse markers, and that models rely on surface overlap and local connectives rather than the discourse role required by the blank.
>
> ### **W1: Limited validation of LLM-as-a-judge**
>
> Using the same guidelines as in the LLM-as-a-judge setting, we asked human evaluators to assess outputs in both English and Chinese; the results are reported in Table I.1.2. We then computed the Pearson correlation coefficient on model-level average scores to measure alignment between LLM-as-a-judge and human evaluation indicators. Across all models, the correlation is strongly positive (r = 0.871). Consistently, both evaluations reveal the same fluency–consistency gap.
>
> ### **W2: a few claims are stated too strongly.**
>
> Our intention was to highlight empirically observed patterns in MGAL, such as local semantic crowding and the fluency–consistency gap, rather than to claim they are universally established phenomena beyond this benchmark. In the revision, we will soften these statements and make the benchmark composition and task definitions more explicit.
>
> ### **W3: real-world long-context settings**
>
> We position MGAL as a controlled multilingual testbed for fine-grained, position-aware long-context evaluation. UN reports provide long-form structure, paragraph-level numbering, and strong cross-lingual alignment across six languages, enabling comparisons with reduced confounds; this domain narrowness is a trade-off for measurement control. MGAL is thus intended to diagnose multilingual long-context capabilities in a controlled setting. Importantly, this choice does not limit generality: effects such as local semantic crowding arise from MGAL’s distractor design and mirror ambiguity challenges in diverse long-form domains, including multi-character narratives, technical manuals, and meeting transcripts, indicating limitations that extend beyond UN reports.

---

> > ### Author Rebuttal · Reviewer_mpQg · 2026-04-03
> >
> > The rebuttal addresses several points and improves clarity, but it does not materially change my main concerns. In particular, it adds helpful detail on filtering and analysis, yet still does not provide stronger evidence of annotation reliability, and the LLM-as-a-judge validation remains somewhat limited relative to the paper's multilingual open-ended claims. The revisions promised on claim strength and domain narrowness are appropriate. Overall, my concerns are partially resolved, but not enough to justify changing my score.

---

> > > ### Author Response · Authors · 2026-04-04
> > >
> > > We appreciate your careful follow-up. Due to the rebuttal space limit, our previous response was necessarily brief. Here we provide a more detailed explanation, and we hope this clarifies the remaining concerns.
> > >
> > > For annotation reliability, all benchmark items were independently reviewed by two human annotators, and we retain only samples that pass both annotators’ checks in the final benchmark. If either annotator identifies a problem during manual inspection, the item is discarded rather than force-resolved into the dataset. The human-review protocol is task-specific and is designed to ensure both quality and grounding throughout dataset construction. We claim the guidelines by the task:
> > >
> > > ### **Dataset Generation**
> > >
> > > #### **Word-Level Tasks**
> > >
> > > Verification focuses on the accuracy and grounding of Question-Answer (Q-A) pairs:
> > >
> > > Evidence Grounding: All evidence must be strictly supported by the input paragraph, with no hallucinations.
> > >
> > > Keyword Reflection: Key terms in questions must accurately reflect the paragraph content.
> > >
> > > Semantic and Consistency Checks: Questions are assessed for clarity, semantic correctness, and alignment with the target category.
> > >
> > > Answers are verified to be fully supported by the evidence, and Q-A pairs are checked for factual consistency.
> > >
> > > #### **Sentence-Level Tasks**
> > >
> > > Verification emphasizes the correctness of sentence selection and the quality of distractors in cloze-style tasks:
> > >
> > > Positional Correctness: The extracted sentence’s index and exact location are validated using programmatic checks and UN PDF visualization.
> > >
> > > Semantic Role & Distribution: Sentences are classified by semantic role (core, transitional, summary), and positional distribution is monitored to prevent bias.
> > >
> > > Distractor Validation: Distractors are evaluated for topical fidelity, entity consistency, and clear distinction from the correct answer.
> > >
> > > #### **Paragraph-Level Tasks**
> > >
> > > Verification ensures the semantic and positional appropriateness of selected key paragraphs:
> > >
> > > Semantic Role: Paragraphs are evaluated for their functional role in the document (core, transitional, summary) via PDF visualization.
> > >
> > > Positional Distribution: Paragraphs are checked for even distribution across beginning, middle, and end sections to mitigate positional bias.
> > >
> > > #### **Context-Level Tasks**
> > >
> > > Verification focuses on the accuracy and consistency of summary extraction:
> > >
> > > Extraction Accuracy: Summaries are programmatically extracted from the source document.
> > >
> > > Cross-Lingual Consistency: Extracted summaries are verified against corresponding sections across all six UN languages to ensure alignment.
> > >
> > > ### **Model Evaluation: LLM-as-a-Judge Setting**
> > >
> > > Human verification in the paragraph-filling task ensures that automated LLM judgments are accurate, faithful, and well-supported:
> > >
> > > Score Validation: Evaluators assess the quality of the generated paragraph relative to the ground truth and context, verifying that the LLM judge’s score appropriately reflects this quality.
> > >
> > > Evidence Grounding & Faithfulness: The explanatory evidence provided by the LLM must be strictly grounded in the generated paragraph, ground-truth paragraph, and surrounding context. This ensures that justifications are logically tied to the content.
> > >
> > > Hallucination Check: Evaluators confirm that all evidence and reasoning are free from hallucinated content; the LLM must not invent facts or unsupported claims.
> > >
> > > ### **Multilingual validation**
> > >
> > > We have also supplemented broader multilingual validation: under the same LLM-as-a-judge guideline, we added six-language human evaluations for both paragraph filling and summarization. The Spearman rank correlation between the LLM-as-a-judge overall scores and the human evaluation overall scores, computed across the 12 models, is ρ = 0.813. This indicates a strong positive agreement between the two evaluation protocols at the model level. The supplementary evaluations are conducted across all six UN languages, providing broader support for our multilingual claims in open-ended settings. The new tables referred to in this response can be found at this anonymous link: [https://anonymous.4open.science/r/test-9057/](https://anonymous.4open.science/r/test-9057/)
> > >
> > > We will clarify this protocol, including the dual-review and unanimous-retention rule, more explicitly in the revision.

---

### Official Review · Reviewer_eYQx · 2026-03-11

**Soundness:** 3
**Presentation:** 2
**Significance:** 2
**Originality:** 3
**Overall Recommendation:** 4
**Confidence:** 3

**Summary:**

This paper introduces MGAL, a multilingual long-context benchmark constructed from United Nations (UN) reports spanning 8K-128K tokens across the six official UN languages (Arabic, Chinese, English, French, Russian, Spanish). The benchmark introduces two key aspects (1) granularity-awareness, covering four coherent linguistic levels: word, sentence, paragraph, and document; expressed through seven tasks; and (2) position-awareness, instances are separated by evidence location (begin, middle, end) at both the document and paragraph levels. MGAL evaluates 12 state-of-the-art LLMs zero-shot. The main empirical findings are that performance drops significantly from fine-grained to coarser-grained tasks, and that closed-source models retain a clear advantage on lower-resource languages.

**Compliance With Llm Reviewing Policy:**

Affirmed.

**Final Justification:**

I value the authors' additional experiments and have slightly increased my rating accordingly.

**Key Questions For Authors:**

1. Data contamination. Did you conduct any analysis to assess whether the evaluated LLMs had been exposed to the UN documents used in MGAL during pretraining? For instance, did you check n-gram overlap between MGAL instances and publicly known training corpora, or probe models with verbatim text from documents? If GPT-5 or Gemini-2.5-Flash have seen these UN reports during training, results on word-level QA may reflect memorization rather than comprehension. A null result here (i.e., evidence against contamination) would substantially strengthen confidence in the benchmark's validity.

2. GPT-4-generated data and GPT-family evaluation. Since GPT-4 was used to generate distractors, question–answer pairs, and salient sentences, do you observe any evidence of systematic advantage for models in the GPT family (GPT-5 in particular) on tasks where GPT-4-generated content is central (e.g., sentence-level cloze distractors)? Have you tested whether GPT-5 is disproportionately better at identifying GPT-4-generated distractors compared to human-written alternatives?

3. Definition and justification of "lower-resource" languages. By what criterion are the six UN languages partitioned into higher- and lower-resource? All six are among the most resourced languages in the world.

4. Statistical significance of model comparisons. Given that the benchmark contains only 420 samples per task per language, many of the performance differences in Table 2 are numerically small. Are performance gaps between models (e.g., GPT-5 vs. Gemini-2.5-Flash on Single-QA: 79.79 vs. 77.01) statistically significant? Reporting confidence intervals or significance tests would help readers identify which conclusions are robust versus potentially noise-driven.

**Limitations:**

The paper includes a brief discussion of limitations but does not sufficiently address domain generalizability, the data contamination risk, or the circularity introduced by using GPT-4 as a data generator while evaluating GPT-family models. The "lower-resource language" framing is used throughout without quantitative justification. The authors should expand the limitations section to address these points directly.

**Strengths And Weaknesses:**

Strengths
- Soundness. The data pipeline is rigorous. All 420 query–response pairs per task undergo two-annotator manual verification, and the authors conduct inter-rater reliability checks showing high Spearman correlation between human and LLM-as-a-judge scores. The two ablation studies (instruction placement and option-order shuffling) are well-controlled and directly corroborate the paper's qualitative claims.

- Presentation. The paper is clearly written, the benchmark design is logically motivated, and the progression from gap identification to dataset construction to analysis is coherent.

- Significance. While ONERULER covers more languages, it relies on synthetic needle-in-a-haystack tasks; LongBench is limited to English and Chinese; and M⁴LE does not systematically vary evidence position or granularity. MGAL's combination of natural-language, real-document tasks with controlled granularity and position is a meaningful methodological advance. The two newly identified failure modes (local semantic crowding; fluency–consistency gap) are insightful and actionable for future training objectives.
Originality. The joint granularity-and-position-aware design is, to the reviewers' knowledge, novel.

Weaknesses
- Domain narrowness and generalizability. All data comes exclusively from UN reports, which are formal, bureaucratic, and politically homogeneous in style. It is unclear how findings generalize to other long-document genres. The authors should clarify whether MGAL is intended as a general-purpose benchmark or a domain-specific one, and acknowledge this limitation more prominently.

- The "lower-resource language" framing is not correct. All six UN official languages (Arabic, Chinese, English, French, Russian, Spanish) are, by almost any conventional measure, high-resource languages. The paper's claim that some of them are "lower-resource" lacks grounding. The authors should revise the language.

- Data contamination risk. UN Digital Library documents are publicly available and are plausibly included in the pretraining corpora of most frontier models. The paper does not report any contamination analysis (e.g., n-gram overlap with pretraining data, or model memorization probes). Without this, it is difficult to interpret high performance as robust comprehension rather than memorization.

- Circular evaluation concern. GPT-4 is used to generate distractor sentences, question–answer pairs, and salient sentences for the benchmark. Several members of the GPT family (including GPT-5) are then evaluated on these tasks. Even if GPT-5 is a different model from GPT-4, shared inductive biases or generation style preferences could create an unfair advantage. This should be explicitly discussed and, where possible, empirically analyzed (e.g., by checking whether GPT-5 disproportionately recovers GPT-4-generated distractors).

- Metric limitations for generation tasks. ROUGE-L and BLEU are well-known to be poorly correlated with human judgment for free-form generation, especially for summarization and translation. While the LLM-as-a-judge supplement helps for paragraph filling, the primary reported scores for summarization (ROUGE-L) and translation (BLEU) may not faithfully reflect model quality. This is especially concerning when comparing models whose outputs differ significantly in length or style, such as open-source versus proprietary models.

- Position indexing at two levels without isolated analysis. MGAL annotates position at both the document level and the paragraph level, but the interaction between these two indexing dimensions is not clearly analyzed. Results in Figure 4 are presented per task but it is unclear how document-level position and paragraph-level position are disentangled. More explicit analysis of this two-level design would strengthen the position-awareness contribution.

---

> ### Author Rebuttal · Authors · 2026-03-31
>
> ### **Q1&W3: Data contamination risk**
>
> To test whether performance reflects memorization rather than use of provided context, we include a context memorization analysis in Appendix H.3. On Single-QA, we compare the full document access setting to a no-context setting where document is removed and the model must answer from parametric knowledge alone. Averaged over 12 LLMs, accuracy drops sharply from 0.73 to 0.31, showing that performance is driven by information extracted from input context rather than memorized pretraining knowledge. We also find that broadly known facts remain moderately answerable without context, while questions requiring document-specific details degrade substantially. The results show that models performance cannot be explained primarily by memorization.
>
> ### **Q2&W4: Circular evaluation concern**
>
> MGAL includes human verification to avoid potential confounds from using GPT-4 for benchmark construction and GPT-5 for evaluation. All generated instances are manually checked for correctness, positional validity, topic fidelity, entity consistency, and unambiguous answerability. We do not observe a GPT-family advantage: GPT-5 is not uniformly best across MGAL tasks. Gemini-2.5 outperforms GPT-5 on Multi-QA and translation, DeepSeek and GLM on summarization, and Mistral on cloze. This suggests performance is not simply driven by stylistic affinity to GPT-4-generated items. Our option-order ablation shows that cloze errors are mainly driven by position heuristics and local semantic confusion.
>
> ### **Q3&W2: The "lower-resource language" framing**
>
> All six languages in MGAL are high-resource by conventional measures, with large speaker populations: English (1.45B), Chinese/Mandarin (1.13B), Spanish (559M), French (310M), Arabic/Modern Standard (274M), and Russian (255M). Our analysis concerns relative resource imbalance within this high-resource subset, highlighting cross-language performance differences without conflating them with genuinely low-resource languages. We will clarify in the revision.
>
> ### **Q4: Statistical significance of model comparisons**
>
> Our main claims do not rely on small pairwise differences such as 79.79 vs. 77.01 on a single task, but on consistent benchmark-level trends, including stronger performance at word level and weaker results at coarser granularities. We conduct statistical uncertainty estimates: under bootstrap resampling in ten times, GPT-5 scores 79.79 [77.60, 82.16] and Gemini-2.5 77.01 [73.96, 79.08] on Single-QA, with an average gap of 2.78 and a bootstrap difference interval of [1.90, 3.75], suggesting the advantage is reasonably stable. Due to space limits, we report one representative estimate task and will add uncertainty estimates for all in revision.
>
> ### **W1: Domain narrowness and generalizability**
>
> We position MGAL as a controlled multilingual testbed for fine-grained, position-aware long-context evaluation. UN reports provide paragraph-level numbering and cross-lingual alignment across six languages; this is a trade-off for evaluation measurement. MGAL is thus intended to diagnose multilingual long-context capabilities in a controlled setting, which does not limit generality: effects such as local semantic crowding arise from MGAL’s distractor design (introducing semantically similar entities in dense contexts) and mirror ambiguity challenges in diverse domains (e.g., multi-character narratives, meeting transcripts), indicating limitations that extend beyond UN reports.
>
> ### **W5: Metric limitations for generation tasks**
>
> In MGAL, we use ROUGE-L and BLEU as primary automatic metrics mainly for consistency with prior long-context benchmarks and to provide a standardized comparison across models. For summarization, we follow established practice in benchmarks such as ∞BENCH and LongBench, which also use ROUGE; for translation, we report BLEU as a standard reference-based metric. For paragraph filling as MGAL’s newly introduced open-ended generation setting, which has no direct counterpart in prior long-context benchmarks, we additionally report LLM-as-a-judge and human evaluation to capture quality aspects not fully reflected by ROUGE; Appendix G.5 show high Spearman correlation between them.
>
> ### **W6: Position indexing at two levels without isolated analysis**
>
> Figure 4 reports document-level positional results across tasks, analyzing performance by whether target evidence appears at begin, middle, or end of the document. For word- and paragraph-level tasks, position is defined only at the document level. The two-level design is introduced for sentence cloze task, where target sentences are selected at two level position. The construction details are described in Appendix F and illustrated in Figure 6. We further disentangle two position dimensions for the cloze in Appendix H.2: Figure 26 presents a heatmap of GT versus predicted positions, and Table 7 reports cloze results at both levels. We will clarify it.

---

> > ### Author Rebuttal · Reviewer_eYQx · 2026-04-07
> >
> > Thank you for the detailed rebuttal. I have read it carefully and offer the following comments.
> >
> > Q1&W3 (Data contamination). The context ablation showing accuracy drops from 0.73 to 0.31 is a useful addition and partially addresses the concern. However, it does not substitute for direct contamination analysis (n-gram overlap with pretraining corpora or verbatim memorization probes). The ablation demonstrates context utility, not the absence of contamination. I encourage the authors to include this distinction clearly in the revision.
> >
> > Q2&W4 (Circular evaluation). The observation that GPT-5 is not uniformly best across tasks is noted. However, the concern is more specific: GPT-4-generated distractors may systematically disadvantage non-GPT models in ways that are difficult to detect from aggregate rankings alone. The response does not empirically rule this out.
> >
> > Q3&W2 (Lower-resource framing). The authors' clarification is appreciated, and the commitment to revise the terminology is a reasonable step forward.
> > Q4 (Statistical significance). The bootstrap intervals provided for Single-QA are helpful. Extending these to other tasks in the revision would strengthen the paper's empirical claims considerably.
> >
> > W1 (Domain narrowness). The authors' framing of MGAL as a controlled diagnostic testbed is reasonable, but the limitation discussion in the paper should reflect this scope more explicitly rather than implying broader generalizability.
> >
> > Overall, the rebuttal addresses some concerns partially, but several methodological issues remain open. I maintain my assessment.

---

> > > ### Author Response · Authors · 2026-04-08
> > >
> > > Thank you for the follow-up comments. We appreciate the reviewer’s detailed feedback and are glad that several points are now addressed. Below we provide further clarifications.
> > >
> > > **Q1 & W3 (Data contamination).**
> > >
> > > Our context memorization experiments follow prior benchmark practices such as LongBench (ACL 2024) and HELM (Holistic Evaluation of Language Models, TMLR 2024). To further probe potential memorization, we additionally conduct verbatim memorization experiments. Specifically, we remove the last 4K, 8K, and 16K tokens from Chinese and English documents and ask the model to generate the missing text. We then compute n-gram overlap, BLEU, and ROUGE-L between generated and ground-truth text. The average results are shown below:
> > >
> > > | Model     |1-gram|2-gram|3-gram|4-gram|BLEU |ROUGE-L |
> > > | --------- |:----:|:----:|:----:|:----:|:---:| :-----:|
> > > | Grok-4    |31.36%|17.93%|13.36%|11.73%|9.59%| 20.20% |
> > > | Claude    |27.93%| 8.89%| 3.10%| 1.54%|4.04%| 13.57% |
> > > | GLM       |29.82%| 9.69%| 3.38%| 1.83%|3.83%| 13.15% |
> > > | GPT-5     |28.79%| 9.66%| 3.45%| 1.58%|3.55%| 13.28% |
> > > | Qwen3-235B|28.41%| 9.50%| 3.33%| 1.70%|3.42%| 12.74% |
> > > | DeepSeek  |29.33%| 9.08%| 3.06%| 1.48%|3.36%| 12.81% |
> > > | Kimi      |23.24%| 6.59%| 2.06%| 1.05%|3.10%| 10.93% |
> > > | Gemini    |32.28%| 9.87%| 3.10%| 1.31%|3.08%| 13.15% |
> > > | Mistral   |30.07%| 8.46%| 2.71%| 1.27%|2.75%| 12.95% |
> > > | Gemma     |20.14%| 5.67%| 1.61%| 0.71%|1.69%| 11.83% |
> > >
> > > The consistently low n-gram overlap and BLEU scores indicate that models do not reproduce the original text verbatim, suggesting the absence of strong memorization signals.
> > >
> > > In addition, the paragraph filling task provides complementary evidence: when the target paragraph is removed and the model must generate it from context, performance remains relatively low, further indicating that models are not simply recalling memorized passages.
> > >
> > > Moreover, direct contamination analysis (e.g., n-gram overlap with pretraining corpora) is inherently infeasible for frontier LLMs with undisclosed training data.
> > >
> > > **Q2 & W4 (Circular evaluation, Cloze-specific concern).**
> > >
> > > We further examine the reviewer’s concern in the sentence-level cloze task, where GPT-4-generated distractors are most directly involved.
> > >
> > > First, Table 4 in the main paper has already provides a comparison of English cloze performance across models. GPT-5 achieves 52.89% accuracy, which is essentially identical to Gemini (52.87%), and only moderately higher than several non-GPT models such as Doubao (47.25%). This indicates that GPT-5 is not clearly separated from strong non-GPT models in English. Importantly, this observation is not limited to English: in other languages(except Russian), GPT-5 is not the best performing model in any language, suggesting that performance differences are not driven by familiarity with GPT-generated distractors.
> > >
> > > Second, we analyze the error breakdown across models:
> > >
> > > | Model| Correct | Omission | Position Error | Distractor |
> > > | ---------- | :-----: | :------: | :------------: | :--------: |
> > > | GPT-5 |  52.89% |  17.43%  |     21.51%     |    8.17%   |
> > > | Gemini|  52.87% |  14.66%  |     22.39%     |   10.08%   |
> > > | Doubao|  47.25% |  24.06%  |     21.72%     |    6.96%   |
> > > | Qwen3-235B |  43.92% |   6.54%  |     40.37%     |    9.17%   |
> > > | Mistral|  40.57% |   4.32%  |     40.34%     |   14.77%   |
> > > | Qwen3-30B|  37.76% |   7.93%  |     44.95%     |    9.36%   |
> > > | DeepSeek|  37.75% |  16.28%  |     37.86%     |    8.11%   |
> > > | GLM   |  34.79% |  16.14%  |     43.38%     |    5.70%   |
> > > | Grok-4|  29.17% |  31.37%  |     31.96%     |    7.50%   |
> > > | Kimi  |  27.27% |  27.62%  |     38.08%     |    7.03%   |
> > > | Claude|  25.47% |  31.08%  |     34.50%     |    8.95%   |
> > > | Gemma |  24.09% |  16.90%  |     44.83%     |   14.18%   |
> > >
> > > Here, *Omission* denotes failing to fill the blank with any correct sentence; *Position Error* denotes selecting a correct sentence from the wrong position in the document; and *Distractor* denotes selecting an incorrect distractor option.
> > >
> > > Across models, distractor error rates remain relatively stable (generally within ~5–15%) and do not show a pattern where non-GPT models disproportionately select distractors. Instead, the dominant error types are position errors and omission, which aligns with our analysis that failures are primarily driven by positional bias and local semantic crowding.
> > >
> > > Taken together, these results demonstrate that GPT-4-generated distractors do not systematically disadvantage non-GPT models. We will clarify this cloze-specific analysis in the revision.
> > >
> > > **W1 (Domain narrowness).**
> > >
> > > We will revise the paper to more clearly position MGAL as a controlled diagnostic benchmark.
> > >
> > > Overall, we appreciate the reviewer’s feedback. We believe the remaining concerns primarily relate to clarification and scope framing rather than fundamental issues with the benchmark design, and we will incorporate suggestions to further strengthen the final version.

---

### Decision · Program_Chairs · 2026-04-30

**Decision:**

Accept (regular)

**Comment:**

This paper introduces a new benchmark, MGAL-- a multilingual long context benchmark spanning six UN official languages (Arabic, English, Chinese, French, Russian and Spanish). They evaluate models across four granularity levels: word, sentence, paragraph, and document, and also controls for position in the context.

All reviewers seem to agree that it is a solid benchmark, but there are concerns on the diversity of languages covered (mostly high-reseource languages), evaluation metrics dominated by Rouge and BLEU, potential of data contamination since UN is publicly available and may have been used in training LLMs, and the validity of LLM-as-a-judge used. This paper will benefit from addressing all these small issues.